# Template Synthesis of Porous Ceria-Based Catalysts for Environmental Application

**DOI:** 10.3390/molecules25184242

**Published:** 2020-09-16

**Authors:** Igor Yu. Kaplin, Ekaterina S. Lokteva, Elena V. Golubina, Valery V. Lunin

**Affiliations:** Chemistry Department, Lomonosov Moscow State University, Leninskie Gory 1/3, 119991 Moscow, Russia; kaplinigormsu@gmail.com (I.Yu.K.); golubina@mail.ru (E.V.G.); VVLunin@rambler.ru (V.V.L.)

**Keywords:** CeO_2_-based catalysts, template synthesis, biomorphic materials, porous oxides, environmental catalysis

## Abstract

Porous oxide materials are widely used in environmental catalysis owing to their outstanding properties such as high specific surface area, enhanced mass transport and diffusion, and accessibility of active sites. Oxides of metals with variable oxidation state such as ceria and double oxides based on ceria also provide high oxygen storage capacity which is important in a huge number of oxidation processes. The outstanding progress in the development of hierarchically organized porous oxide catalysts relates to the use of template synthetic methods. Single and mixed oxides with enhanced porous structure can serve both as supports for the catalysts of different nature and active components for catalytic oxidation of volatile organic compounds, soot particles and other environmentally dangerous components of exhaust gases, in hydrocarbons reforming, water gas shift reaction and photocatalytic transformations. This review highlights the recent progress in synthetic strategies using different types of templates (artificial and biological, hard and soft), including combined ones, in the preparation of single and mixed oxide catalysts based on ceria, and provides examples of their application in the main areas of environmental catalysis.

## 1. Introduction

Porous oxide materials with the unique conducting and textural properties and variable morphology are applied in many fields, such as energetics, electronics, separation processes, biotechnology, and catalysis. Among various metal oxides, porous cerium dioxide (ceria, CeO_2_) and complex ceria-based systems are the most commonly used as catalysts or as non-inert supports for a wide range of catalytic reactions. These materials are distinguished by their relatively low cost and toxicity, high thermal and chemical stability, and unique physicochemical properties related to the chemical composition and fluorite crystal structure: facile transfer between cerium oxidation states, high concentration of oxygen vacancies and other defects, high oxygen storage capacity and oxygen ion conductivity [1]. Therefore, they are extensively used in various fields of environmental catalysis such as three-way catalysis (TWC) [2,3], removal of SO_x_ in fluid catalytic cracking processes [3], dehalogenation [4], dehydrogenation [5], and partial hydrogenation processes [6], oxidation [7], NO_x_ reduction [8], hydrocarbon reforming [9], thermochemical water splitting using concentrated solar energy combined with subsequent production of H_2_, syngas, or hydrocarbons [10], hydrocarbons production by CO_2_ hydrogenation [11] etc.

The catalytic properties of CeO_2_ can be improved by several ways, which can be grouped as follows: (i) introducing doping agents and other promoters [12], and (ii) using various synthetic methods [13] that allow developing ceria porous system with a defined texture and morphology. The morphology and porous structure of the catalyst are essential factors which determine the overall specific surface area, particle shape and size, mass transport and diffusion parameters of the process, surface centers for anchoring of dopant, such as metal nanoparticles on oxide surface, and accessibility of active catalytic sites (the fraction of highly active facets, edges, and corners containing coordinatively unsaturated ions and surface defects) [14]. As it has been demonstrated in recent reviews, the morphology of pristine and doped ceria catalysts plays a crucial role in the CO oxidation [15] and in the VOCs catalytic combustion [16]: the particles of well-defined geometry (nanorods, cubes, etc.) provide preferable crystal planes with improved oxygen storage capacity and other properties important for oxygen and substrate activation.

The development of methods for the controlled synthesis of ordered porous materials started in the middle of the twentieth century and is still continuing. The template-assisted synthesis of oxide systems is one of the most successful and fruitful methods for catalyst preparation as it provides a plenty of advantages. It allows producing materials with the desired meso-, macroporous or even hierarchical porous structure and improved texture and morphology. Moreover, these properties can be tuned by the proper selection of the template, by combining several templates, and by varying of the preparation conditions. The templated methods are especially efficient for the preparation of oxide catalysts for the applications in which the texture properties and morphology are crucially important to provide access of large reactant molecules or their aggregates to the active centers of the surface, e.g., soot oxidation [17], processing of biomass [18], etc.

Many reviews and even books devoted to porous oxide materials, including cerium dioxide were published over the past decade [15,16,19,20,21]. However, these works either cover a limited range of catalytic systems and processes, or do not fully consider the whole spectrum of template materials that could be used to synthesize ceria-based catalysts. Besides, new works have recently emerged in this area. In this review, we try to summarize and analyze the most recent reports published in the last decade on the template synthesis and catalytic applications of ceria-based materials with defined porous structure. This article is organized as follows: firstly, a discussion of the basic concepts of template synthesis is provided, general synthetic pathways are listed, and a classification of different templates used for the preparation of CeO_2_-based catalysts is proposed. Then, various preparation methods for cerium oxide systems based on artificial templates and their application in the environmental catalysis are described in detail. After that, we highlight the “green” synthetic procedures using biomaterials as templates to produce effective biomorphic ceria-based catalytic systems. In the conclusion section our personal assessment of the trends and prospective directions in which scientific work in this field can be focused in the future is presented.

## 2. General Strategies for Template Synthesis of Porous Catalysts Based on CeO_2_

Many different classifications of template methods are presented in the literature. However, in almost all template methods the synthesis of a porous oxide material comprises two essential stages: (i) the arrangement of a solid oxide precursor (or precursors) around a template cluster or matrix, and then (ii) producing of the oxide material with a porous structure by partial or complete template removal. Templates can be divided into two large groups according to their nature (artificial and biotemplates), as shown in Figure 1. Some carbon materials prepared from biomaterials can be attributed to both groups, because, on the one hand, they are obtained artificially, but on the other hand their morphology represents the original biomass. In addition, in some syntheses, the carbonization of the biotemplate precedes its removal.

Artificial templating agents/precursors can be based on pure organics and polymers [22,23,24,25,26,27,28,29,30,31,32,33], silicon-organic compounds and mesoporous silica [27,34,35,36,37,38,39,40,41], metal-organic frameworks [42,43], synthetic carbon materials [44,45,46,47], etc.

Biological templates are a large group of biomass-based materials that are used in raw [48,49,50,51,52,53,54,55,56,57] and processed form, e.g., textile [58,59,60,61,62,63]. Both plant, animal and microorganism biomass are used for this purpose.

The preparation methods of porous oxide systems from artificial materials can be further divided in many subgroups primarily due to the fact that a chemical reaction generally takes place in the mixed solution or suspension of complex composition helping in homogeneous distribution of an oxide precursor over template matrix pre-made or self-organized in the reaction mixture. Moreover, these ways of the template structure formation could be applied both separately and in combination. Complex strategies combining different templating techniques often help producing more complicated highly ordered or hierarchical morphology [37,64,65,66].

The synthetic approaches based on biological templates look technically simpler—in this case the infiltration/impregnation of a biological material with a precursor solution (usually inorganic/organic salts or complexes) is followed by calcination or dissolution of a template [21]. However, due to the complex and variable composition of natural materials the properties of the resulted oxides can significantly vary. The main factors that affect the choice of the template and preparation procedure are the desired texture and morphology of the oxide, stability of the prepared structure to thermal treatment during preparation and use, chemical resistance in the catalytic reaction conditions, availability and cost of template and oxide precursors, solvents and other materials, simplicity and reproducibility of the synthesis, and the compliance of each stage and the entire synthesis with the standards of green chemistry [21].

These preparation methods, the main characteristics of the synthesized ceria-based materials and their application in various catalytic reactions will be described in detail in this review.

## 3. Ceria Preparation Methods Based on Artificial Templates

A possible classification of the synthetic strategies for the production of CeO_2_-based porous materials using artificial templates is shown in Figure 2. It comprises four major categories: well-established (i) soft- and (ii) hard-template methods; (iii) relatively new self-template methods, and (iv) complex synthetic procedures, combining two and more templates. The synthesis often includes additional steps such as hydrothermal treatment, and the use of auxiliary reagents e.g., agents for swelling, pH control, complexation of ceria precursor with template etc.

In this part the classical preparation methods using soft and hard templates will be considered. The basic concept of each synthetic strategy will be described and illustrated by several recently published examples.

A comprehensive review of the methods of 1D ceria synthesis is presented in [67]. The use of both soft (octadecylamine, dodecylsulfate, polyethylene glycol, quaternary ammonium salts) and hard (anodic alumina membranes, carbon nanotubes, mesoporous silicas with hexagonal and cubic symmetries, KIT-6) templates for the synthesis of mainly ceria nanowires, but also lamellar sheets and nanotubes is elucidated in this work. Although this article is mainly aimed at highlighting the use of such structures in biological processes (disease treatment, bioimaging and drug delivery), the produced materials are certainly promising in the field of catalysis.

### 3.1. Soft Template Methods

The soft-template routes result in the direct formation of porous structure by assembly of precursor molecules on “soft materials” with non-rigid shapes. There are plenty of materials which can be used as templates, such as organic macromolecules, ionic liquids, and surfactants. These materials can assemble into supramolecular aggregates forming flexible matrix for micro-/meso-/macrostructures. Varying the composition of the reaction medium and such parameters as temperature and pressure during formation of the soft template structure provides an easy way for controllable tuning of the chemical composition, structure, morphology, size, orientation, pore distribution, and internal and external surface properties of the synthesized porous materials. Some authors [68] use the name “endo-templating” for methods involving soft templates because in this case precursor molecules surround aggregates of template molecules and form solid ceria phase outside the template structure (template is situated inside the ceria precursor, inside = “endo”). The voids remaining in the material after removal of the template form a porous system. It is important to note that there is a confusion in literature about the name of these methods. Several authors [68,69] call them as “exo-templating” (the precursor is outside the matrix, outside = “exo”). Further in this review, we will stick to the former name. It seems that the classification of soft materials only as “endo”-templates is not quite correct, because there are direct or inverse micelle assembly mechanisms resulting in the formation of endo- or exo-structures, respectively.

#### 3.1.1. Ionic Surfactants as Soft Templates

Micelles and vesicles formed by amphiphilic molecules when their concentrations exceed critical values are widely used as soft templates. Their packing pattern can be easily controlled by varying the concentration of surfactant, ionic strength, temperature, pH values, or by introduction of additives, which allows producing structures with different shapes including spherical, cylindrical, and bilayer spherical aggregates [70]. Amphiphilic compounds (surfactants and polymer materials) can be classified into three groups: cationic, anionic, and non-ionic.

A striking example of widely used cationic surfactants is alkyltrimethylammonium salts, especially cetyltrimethylammonium bromide (CTAB). A lot of papers report the use of CTAB to design novel high surface area ceria-based materials for applications in catalysis including methane dry reforming [22], chlorobenzene destruction [26], toluene oxidation [71], solvent-free oxidation of benzyl alcohol [28], transesterification of ethylene carbonate (EC) with methanol to dimethyl carbonate (DMC) and other cyclic carbonates and alcohols [29] etc. Other cationic templates have also been applied in the field of catalysis, but they are not so popular as CTAB and its counterparts. Summarizing the results of plenty of works, CTAB template creates a mesoporous structure of ceria with the narrow pore size distribution. It also improves S_BET_, reducibility of cerium, oxygen mobility, and can influence the acid-basic properties, which is important for adsorption of reactants and in acid-base catalysis.

A series of cobalt and/or iron-modified ceria catalysts for diesel soot combustion was synthesized using CTAB-assisted co-precipitation method [23]. It was shown that all prepared systems had mesoporous texture and monomodal pore size distribution. The specific surface area of the ternary metal oxides was relatively large (85–115 m^2^/g). Interestingly, the specific surface area of co-doped oxides increases with increasing the iron loading. The Co-Fe/CeO_2_ systems were especially efficient in the soot oxidation under both tight and loose contact conditions because of their unique mesoporous structure and additional oxygen vacancies on the surface. The large surface area and high porosity of the catalyst are known to increase the number of contacts between the catalyst and soot particles, thereby improving the catalyst activity in the soot oxidation.

The double oxide CeZrO_x_ (CZ) materials prepared using the CTAB template or by precipitation with urea were compared in [30]. The template method provided higher S_BET_ value for the samples with high Zr content, in contrast to the precipitation with urea, in which the highest specific surface areas were observed for pure oxides and at Ce:Zr = 1. In the case of CZ, the influence of Ce:Zr ratio on the texture, structure, morphology, and surface properties is more complex than in the case of CeO_2_-TiO_2_ composite. When CTAB is added, the concentration of mesopores increases, with their size distribution and shape being dependent on the Zr modifier loading. The high catalytic activity and selectivity to ethyl acetate total oxidation is explained in this work by the improved Lewis acidity and reduction ability of materials with the relatively high extent of lattice and surface defects formed for three reasons: as a result of smaller crystal size, on the crystal planes with higher Miller indices, and by incorporating zirconium ions into the CeO_2_ lattice and vice versa.

The evaporation-induced self-assembly (EISA) is a widely used method for the preparation of ceria-based oxides [27,31,72]. In this process the self-association of individual components into an organized structure is stimulated by the slow solvent evaporation. Thus, in our scientific group CTAB and citric acid were used as a template and complexing agent, respectively, to synthesize Ce_0.8_Zr_0.2_O_2_ (CZ) and MnO_x_-Ce_0.8_Zr_0.2_O_2_ (Mn-CZ) oxide catalysts by EISA method (Figure 3). These catalytic systems were tested in CO oxidation [31]. It was noticed that the catalytic action depended on the way manganese was added to ceria-zirconia oxide: during the self-assembly of oxide precursors on a micelle template, as in “one-pot” method used to prepare Mn-CZ, or by post-impregnation of the CZ prepared by EISA method with a manganese precursor followed by calcination to remove the CTAB template (Mn-CZ IM). Both ways led to a substantial decrease in S_BET_ (by about half) compared to the non-modified Ce-Zr sample for which S_BET_ was relatively high (83 m^2^/g). Such deterioration of porosity could be explained by the poor ability of manganese ions to form stable complexes with citric acid. The lowest S_BET_ value of MnO_2_ prepared by EISA (20 m^2^/g) confirmed this assumption. In both modified samples manganese oxides were partially distributed on the surface and partially sealed inside CZ particles. However, electron paramagnetic resonance study of Mn-CZ revealed that only the minor fraction of Mn^n+^ ions are incorporated or intercalated in the bulk or in the subsurface layer of CZ oxide crystal lattice, which confirmed the triple oxide formation. No traces of such ions were found in Mn-CZ IM. Interestingly, despite the low specific surface area, the ternary oxide system prepared with impregnation step exhibited higher CO conversions values in the whole studied temperature range (100–450 °C) than CZ and Mn-CZ. This fact can be explained by the mosaic surface structure formed by alternating surface areas enriched with CZ and MnO_x_ that can provide the additional adsorption sites on the surface of Mn-CZ IM sample. Thus, it was clearly demonstrated in our work that the method of adding the third component to cerium-zirconium systems has a significant effect on the catalytic properties.

The modification of cerium oxide with nickel makes it possible to produce catalysts for various processes. Thus, binary nickel-cerium oxide materials were synthesized by the soft-template method using CTAB as a template and modified with nickel using two techniques: the “one-pot” method where ceria and nickel oxide precursors were co-precipitated in the presence of CTAB, providing high S_BET_ values of 170–210 m^2^/g, and the post-impregnation by depositing Ni on the soft-templated ceria support through incipient wetness impregnation, in which case S_BET_ was slightly lower (155–170 m^2^/g). The catalytic tests were performed in CO_2_ methanation [33] after mild H_2_ treatment (400 °C). Both series were found to be highly active and selective towards methanation. The catalytic results were explained in this work in terms of the CO_2_ and H_2_ activation on different phases. NiO nanocrystals of about 4 nm in size were identified in the “one-pot” samples, regardless of the Ni loading. According to the author’s explanation, such small nanocrystals did not sinter due to the strong metal-support interaction. Nickel deposition by the impregnation led to larger NiO particles (about 20 nm), and they agglomerated during reduction. Interestingly, despite the significantly different NiO crystal size, comparable CO_2_ conversion values were observed for all catalysts, but “one-pot” samples exhibited superior performance at increased space velocities (CO_2_ conversion was 50 and 3 mol.% over the “one-pot” and impregnated catalysts with the same 1.5 Ni:Ce molar ratio). The authors underlined the important role of the highly uncoordinated Ni atoms at the metal-support interface, the number of which is higher in small Ni crystallites. These atoms are responsible for hydrogenation of CO_2_ species activated on nearby ceria sites.

Similar study was performed in [32], where the series of NiO/CeO_2_-ZrO_2_ mixed oxides with the same Ni content and different Ce:Zr molar ratios were also prepared by the “one-pot” CTAB-templated method. In this work, a mixture of carbon oxides was subjected to methanation. In all ceria-containing samples the Ni^0^ particle size was the same, about 6 nm. Ternary oxides demonstrated remarkably high S_BET_ (200 m^2^/g and more) and provided complete CO methanation, whereas CO_2_ conversion was much lower and increased with Ce content, at least up to Ce:Zr = 1. Simultaneously, the specific surface areas decreased. The NiO/CeO_2_-ZrO_2_ sample with Ce:Zr = 1 was stable during at least 50 h time-on-stream. The authors explained the beneficial effect of the Ce content by the increased NiO reducibility and the higher ability of CeO_2_ to adsorb and activate CO_2_. However, at high Ce:Zr ratio the larger amount of activated hydrogen would favor the reverse methane dry reforming reaction producing CO_2_. As a result of these balanced processes the overall CO_2_ concentration remained almost constant.

Ceria-titania composites with 2:8, 5:5 or 8:2 Ce:Ti molar ratios were synthesized using the CTAB template with the subsequent hydrothermal treatment and calcination at 600 °C [73]. They were tested in methanol decomposition to produce hydrogen as a potential alternative fuel, and ethyl acetate oxidation to check their ability in VOCs disposal. These oxide materials were combined to overcome their individual disadvantages: wide bandgap of TiO_2_ and poor thermostability of CeO_2_. No mixed oxide formation was found by Raman spectroscopy and XRD, but according to XPS the intimate contact between individual oxides resulted in the generation of Ce^3+^ and Ti^3+^ ions as well as oxygen vacancies. The increase of the temperature of hydrothermal treatment weakened the contact between individual oxides. The highest S_BET_ values were observed for the composites with Ce:Ti = 2:8 hydrothermally threated at 100 °C (166 m^2^/g) and 140 °C (127 m^2^/g) with predominantly cylindrical pores. The increase in Ce content led to the changes in pore shape from the “cage-like” at Ce:Ti = 5:5 to “slit-like” at 8:2 with the simultaneous deterioration of homogeneity of mesopore size distribution. The best texture was achieved at low Ce:Ti ratio and low temperature of the hydrothermal treatment. This material comprised highly dispersed CeO_2_ particles anchored on the oxygen vacancies of TiO_2_, high concentration of Lewis acid centers and mobile oxygen. All these features ensured the high catalytic activity and selectivity in total oxidation of ethyl acetate to CO_2_ and methanol decomposition to syngas. In contrast, bulk ceria crystallites partially substituted with Ti with worse texture parameters are formed at high Ce:Ti ratio, providing higher density of Lewis acid centers, decrease in the concentration of Ti^3+^ and Ce^3+^, deterioration of catalytic activity, and change in selectivity. Ethyl acetate hydrolysis to ethanol, and methanol decomposition to methane were registered.

Skillful use of the template allows synthesizing complex anisotropic structures. In [74] nanosized Au@CeO_2_ core-shell catalyst was produced by controlled hydrolysis of cerium acetate precursor in the presence of CTAB as a soft template. The template hinders hydrolysis of the ceria precursor to produce mushroom-like structure in which golden nanorods are half-covered with CeO_2_, whereas without the template metal nanorods are completely covered with the ceria shell. Anisotropic structure of the catalyst comprising golden nanorods with the specified aspect ratio partially covered with ceria provides appropriate activity in photocatalytic reduction of 4-nitrophenol under near-infrared laser irradiation due to improved plasmon absorption.

Among different surfactants (CTAB, sodium dodecyl sulfate, dodecyltrimethylammonium bromide, cetyltrimethylammonium chloride, polyvinylpyrrolidone, and KBr) only CTAB one led to the desired mushroom-like morphology [74]. It seems that bromide ions play an important role during the anisotropic growth, possibly due to stronger (compared to Cl^−^) interaction with noble metals, affecting the growth of surface nanostructures.

Summarizing the data on the ceria systems prepared using the CTAB template, and modified with manganese, Co-Fe or nickel dopants, the following conclusions can be drawn:Template can significantly improve the textural properties of both unmodified and modified ceria, but the careful choice of dopant is needed, because some modifiers can hinder the pore structure formation in synthesized material. For instance, the one-step CTAB-templated method results in the formation of the Ni/CeO_2_ and Ni/CeZrO_x_ (Ni/CZ) oxide systems active in CO_2_ methanation while the similar technique used for modification of CeZrO_x_ with Mn in [31] did not lead to highly effective catalysts for CO oxidation;A well-developed porous structure is a beneficial quality for a heterogeneous catalyst, but many other factors may outweigh its influence on catalytic properties. Nickel particle size [33] or the degree of supported nickel reduction [32] were proposed as the key factors, which determine the catalytic action of Ni/CZ in the methanation of carbon oxides, but tuning of reaction conditions and composition of reaction mixture provide the way to achieve the desirable values of these parameters;The nature of cation and anion in polar templates can play significant role during ceramic synthesis. Thus, the presence of residual anions in the oxides prepared by CTAB-assisted method can affect the surface morphology and therefore catalytic activity. However, the degree of exposure to such ions is difficult to predict, since the effect depends on their concentration, nature of template, preparation conditions, catalyst composition, and type of catalytic reaction.

Anionic surfactants form another group of soft-templates with the long hydrocarbon tail combined with the negatively charged “head-group”. A prominent representative of this class of amphiphilic compounds is sodium dodecyl sulfate (SDS), which is widely used for the synthesis of different oxides (TiO_2_, SiO_2_, zinc oxide, etc.), but there are only scarce references about the use of this or other anionic surfactants for the synthesis of ceria-based oxides.

Thus, the ceria-zirconia mixed oxides were prepared using SDS via the sol-gel route in [24]. After calcination at 500 °C they exhibited regular pore structure and the appropriate values of specific surface area of about 100 m^2^/g. The authors believed that these materials could be successfully tested for various catalytic applications; however, the article lacks the data on the catalytic tests. SDS can also play a role of the foaming agent and hydrophobic modifier for ceria NPs on the surface of hard template (silica hollow spheres) [75].

#### 3.1.2. Non-Ionic Surfactants and Polymers as Soft Templates

Another group of soft templates comprises non-ionic surfactants, such as fatty alcohols, esters and ethers with long carbon chain, and block copolymers containing uncharged hydrophilic and hydrophobic moieties. These templates are distinguished by the absence of counterions, which, as mentioned above, can significantly affect the properties of the resulted product.

In [27] the mesoporous ceria catalysts prepared both by EISA method using Pluronic F127 triblock copolymer as a soft template and by nanocasting using SBA-15 mesoporous silica as a hard template were compared in benzene oxidation. The hard-templated ceria showed relatively large specific surface area (92 m^2^/g) and highly defective internal structure, while the soft-templated sample exhibited the morphology of 3D-linked filaments with the low content of internal defects and lower specific surface area of 32 m^2^/g. The hard-templated sample demonstrated the improved benzene oxidation activity with 50% benzene conversion achieved at 257 °C, which was much lower than that for the soft-templated CeO_2_ (384 °C). However, the catalytic properties of soft-templated ceria can be significantly improved by etching with NaOH, which increases the number of surface defects. They can adsorb active oxygen species providing low-temperature benzene oxidation. Thus, a simple modification of the soft template methods can improve the properties of the produced oxides.

Polymer templates from the pluronic series have been widely used for the synthesis of nickel catalysts supported on pure and modified cerium oxide.

Two strategies, the template one using triblock copolymer Pluronic F127 and the polymerizable complex method, were compared for the synthesis of modified ceria [76]. A wide range of modifiers (Gd, La, Mg) were tested to select a suitable support for nickel catalysts. Different ceria precursors were used (chloride and nitrate, respectively), whereas all modifiers were introduced as nitrates in both the preparation strategies. The conditions of thermal treatment of Ce_1−x_M_x_O_y_ systems were varied in a wide range (final temperature was 300, 500 or 800 °C, duration 4 or 24 h, atmosphere of air or reductive mixture of 30% H_2_/Ar). Ni was added by impregnation. Compared to the competitive technique the template method provided the higher specific surface area nearly for all dopants at air calcination temperatures of 300 and 500 °C with exception of pure CeO_2_ calcined at 500 °C. Doping with lanthanum and calcination at 500 °C lead to the highest S_BET_ both for the support and Ni catalyst; to diminish a decrease in the specific surface area due to the Ni addition, the thermal treatment in H_2_ + Ar atmosphere was recommended. The enhancement of the nickel-support interaction, which stabilizes highly dispersed Ni species and improves thermal and cocking stability in autothermal reforming of ethanol at 200–700 °C, was achieved with a high molar fraction of the dopant and a decrease in the calcination temperature, which was demonstrated by the example of La-modified samples.

In [25] the NiO/ZrO_2_-CeO_2_ composites were prepared by the soft-template method using a different triblock copolymer Pluronic P123 (Ce:Zr molar ratio of 9:1). The produced catalyst showed suitable textural and structural properties for the use in catalysis or as an anode in the solid-oxide fuel cell (SOFC). The choice of calcination conditions allowed tuning the crystal size and promoting phase stabilization. The calcination at lower temperature of 400 °C resulted in the larger total pore volume, higher specific surface area and smaller crystallite size (mostly cubic shape) of biphasic NiO/ZrO_2_-CeO_2_, enhancing NiO and CeO_2_ reducibility, while calcination at higher temperature of 540 °C only improved the Ce^4+^ reducibility. Both systems were equally active in methane conversion, but for the sample thermally treated in milder conditions no signs of carbonaceous deposits formation were noticed. This enhancement in the ceria redox properties may be responsible for the improved surface oxygen exchange, allowing the gasification of carbon species and completely preventing carbon deposits formation, which is typical for the catalysts with similar composition but different morphology synthesized by co-precipitation [77].

Similar Pluronic P123 template was used to produce a catalyst comprising niobium oxide confined by ceria nanotubes for the selective catalytic reduction (SCR) of NO_x_ [78]. Interestingly, in this study CeCl_3_·7H_2_O was used as a ceria precursor instead of the most common nitrate one, and ethanol washing for template removal instead of the more common calcination. The produced composite with the nanotube morphology showed better synergistic effect than its counterpart comprising niobium oxide on ceria nanoparticles, demonstrating higher SCR activity and remarkable resistance to potassium, phosphorus, and lead poisons. Indeed, nanotubular catalyst ensured more than 90% NO_x_ conversion in a broad temperature region of 275–450 °C, while nanoparticulate material demonstrated a similar efficiency only in the narrow temperature range around 350 °C. Similarity in the crystal structure and specific surface area of the both materials led the authors to the conclusion about the crucial role of morphology in ensuring the high efficiency of niobium-cerium nanotubular oxide in the SCR of NO_x_.

Tin oxide is another promising modifier due to the ability to form solid solutions with cerium oxide and the relatively low cost as compared to zirconia. A CO oxidation activity of CeSnO_x_ (CS) and CuO_x_/CeSnO_x_ (Cu-CS) catalysts prepared by CTAB or Pluronic P123 was compared in our scientific group [79]. Catalytic properties of prepared systems strongly depend on the template nature and copper modification technique because these parameters determine degree of interaction between different components in Cu–Ce–Sn oxide systems. The combination of CTAB-templated method and “one-pot” copper addition technique led to the more uniform distribution and partial incorporation of copper ions into the CS lattice, which provided high oxygen mobility and defectiveness. These facts explain why Cu-CS CTAB sample (S_BET_ = 84 ± 8 m^2^/g) exhibited excellent catalytic properties over the entire temperature range studied. In contrast, Pluronic 123-templated counterpart (S_BET_ = 96 ± 9 m^2^/g) showed 40–60% conversion of CO only at relatively low temperatures, and it was less effective in the high-temperature range.

Not only the template, but also the nature and concentration of a swelling agent can influence the texture and other properties of ceramics. Thus, the influence of the weight ratio of template (Pluronic P123) and swelling agent (tri-isopropyl-benzene, TIPB) on the properties of porous double oxide Zr_0.1_Ce_0.9_O_2_ as well as 3 and 10 wt.% Ni/CZ catalysts produced by the post-impregnation was studied in [80]. The template was removed by calcination at an unusually low temperature of 400 °C. The produced materials were mainly mesoporous with S_BET_ of about 110 m^2^/g; but the growth of the swelling agent concentration increased microporosity. In this way the gas permeability and S_BET_ that are important characteristics for Intermediate Temperature-Solid Oxide Fuel Cell (IT-SOFC) and catalytic applications can be increased.

Several petrochemical processes, e.g., methane oxidation, can be efficiently performed only at high temperature, at which the sintering of the particles of active component is highly probable. Thus, new strategy was developed for the synthesis of ceria-supported nanorods of the noble metals, confined in the shell of silica [36]. The mixture of templating agent surfactant NP-5 (polyethylene glycol mono-4-nonylphenyl ether), solvent and hexane was used to form the micro-emulsion system. A solution of precursor salts of the noble metal and cerium oxide was added to the organic mixture and the fine Pd-CeO_x_ wires were precipitated with alkali. The silica shell was created by adding a silica precursor, e.g., tetraethoxysilane (TEOS), in the last step. The final Pd-CeNW@SiO_2_ catalyst obtained by calcination at 600 °C showed outstanding stability toward moisture and SO_2_ during methane combustion [36]. The temperature of 100% methane conversion to CO_2_ and H_2_O was the lowest for Pd-CeNW@SiO_2_ (350 °C) compared to Pd@SiO_2_ (375 °C), and the commercial Pd/Al_2_O_3_ catalyst (425 °C), and it decreased even more in the cooling cycle during catalytic test. The Pd-Ce-O_x_ core phase remained isolated even after 50 h time-on-stream at 800 °C. Its Pt-containing counterpart demonstrated very good catalytic properties in CO and toluene oxidation [38]. 2%Pt-CeO_2_NW@SiO_2_ catalyst was stable during 100 h time-on-stream in the oxygen-enriched CO + O_2_ + N_2_ reaction mixture, and the temperature of 50% CO conversion was 75 °C lower than over the 2%Pt-CeO_2_NW/SiO_2_ counterpart simply supported on silica rather than confined into silica shell. TPR-H_2_ data showed that the reduction of both surface and bulk Ce^4+^ proceeded at much higher temperatures than in the case of the common Pt/CeO_2_/SiO_2_ catalyst (the difference was about 75 and 150 °C, respectively), which was attributed to the more intimate contact of ceria and silica. However, the similar effects can be caused by diffusion limitations due to water formation during catalyst interaction with H_2_. The activity of Pd-Ce-O_x_ or Pt-Ce-O_x_ phases resulted from the higher degree of interaction between the metal and cerium oxide, while the stability is ensured by a protective microporous silica shell.

Thus, the use of the soft template methods is a convenient way to synthesize porous ceria-based oxide systems, and the template is easily removable by calcination in air under relatively mild conditions. However, the nature of ceria precursor (e.g., cerium chloride or nitrate) and the presence of additional ions of doping metals (e.g., rare-earth metals, Pd, Pt, Ni, Cu, etc.) in the reaction solution can significantly influence the textural properties of the resulted materials. The interaction of such ions with the template molecules/aggregates can affect hydrolysis and polymerization processes and change the degree of interaction between the modifier and CeO_2_. 

### 3.2. Hard Template Methods

The hard-template synthesis simplifies designing of oxide systems with tunable morphologies, texture, and higher crystallinity. These methods are considered more predictable and controllable than their soft template counterparts. A variety of rigid porous or micro/nanoparticulate carbon and silica materials, such as (micro)spheres, tubes, filaments etc. can be used as hard templates. For a typical synthesis, at least three separated steps are required: (i) template synthesis, (ii) deposition of an oxide precursor on the template, and (iii) template removal. Interaction of “hard materials” with the precursor salts can proceed through both “exo-templating” (template is outside (“exo”-), where matrix pores are filled with precursor, e.g., mesoporous silica, and “endo-templating” using, e.g., polymethylmetacrilate (PMMA) microspheres [68].

#### 3.2.1. Carbon-Based Hard Templates

Carbon nanotubes (CNT) are becoming an increasingly common and inexpensive carbon material, and it is not surprising that they were used as a template for cerium oxide production [44]. In this work they were preliminarily treated with nitric acid to create defects and better anchor the precursor (cerium nitrate) on the CNT surface. Ceria nanotubes were more active in CO oxidation than bulk and even nanodispersed CeO_2_ due to the higher S_BET_ (about 80, 6 and 30 m^2^/g, respectively). It is expected, that the electronic state, crystallinity, and reducibility of these materials also differ, but they were not thoroughly studies in this work.

Not only surface defects produced by acid etching of a carbon material but also carbon nitride species may be responsible for the formation of sites with the improved dopant adsorption. Graphitic carbon nitride (g-C_3_N_4_) served as a hard template to produce the Ce-Ni binary oxide via sol-gel method [47]. The catalyst exhibited the excellent activity and selectivity in CO_2_ methanation despite the low S_BET_ (only 19 m^2^/g after calcination in air at 500 °C for 4 h and reduction by H_2_ at 450 °C for 1 h). It was found that this synthesis method positively affected several catalyst properties: the concentration of oxygen vacancies, the strength of the nickel-support interaction, etc., which favors H_2_ dissociation and CO_2_ adsorption. Moreover, anchoring on g-C_3_N_4_ prevented the nickel particles from migration and sintering until this sacrificial template was burned.

The new preparation strategy using graphene oxide (GO) flakes as a hard template to produce ceria catalysts for CO oxidation and dry reforming of methane (DRM) was reported in [45]. According to the results of the physicochemical study, ceria reproduces the flake morphology of the template leading to higher specific surface area and concentration of oxygen vacancies than non-templated ceria particles prepared by the classical precipitation technique and calcined at the same temperature. Moreover, the flat structure limited the thermal diffusion of surface atoms thus enhancing thermal stability during calcination and in the reaction medium. For these reasons, GO-templated ceria flakes exhibited the improved catalytic properties in CO oxidation. Moreover, Ni-doped ceria flakes demonstrated high resistance to Ni sintering and considerably higher activity in the conversion of methane and carbon dioxide. Significant improvement of long-term stability in DRM tests compared to non-templated Ni/CeO_2_ sample (Figure 4a,b) is explained by the stronger Ni-support interaction. Thus, the use of GO as a sacrificial template offers a potential route for the synthesis of thermally stable ceria catalysts with better performance for high temperature applications such as automotive exhaust control catalysis or DRM.

Copper-ceria nanosheets with the enhanced interaction between components highly desired in the catalysts for CO oxidation were also synthesized using GO as a sacrificial template [46]. The best calcination temperature was found to be 600 °C: the corresponding sample comprising mixed oxide nanosheets about 10 nm thick demonstrated the highest concentration of oxygen vacancies (7.79%) and active copper species sites (2756 μmol/g), and the best efficiency in CO oxidation, compared to the samples calcined at 400, 500 and 700 °C. The complete conversion of CO that was fed with a significant space velocity of 54,000 mL/(h g_cat_) was achieved over this catalyst at 90 °C; note, that the efficiency of the sample calcined at 500 °C was nearly similar. Moreover, these composites were water resistant and stable during 60 h time-on-stream.

The lightweight monolith 3DOM ceria ceramic can also be produced by the impregnation of carbon blocks prepared by pyrolysis of cork in N_2_ at 900 °C with an aqueous solution of cerium nitrate [81]. Unlike other templates with the wood morphology, open pores prevailed in this material. The authors demonstrated the influence of the template impregnation mode with the precursor solution (from one to four cycles), and pressure (6000 Pa or 20 MPa) on the density, S_BET_ and porosity of synthesized ceria blocks. For example, low pressure favored higher porosity, suitable for the use of such blocks in catalysis. However, calcination temperature must be high (about 1600 °C) to achieve mechanical strength sufficient for easy handling. The cited work does not provide data on S_BET_ and pore sizes; taking into account the high calcination temperature and the morphology in the SEM images, very low meso- and micropore contribution can be assumed.

It can be concluded that the reproduction of the morphology of carbon-containing nanomaterials (nanotubes, nanoflakes, 3DOM-structures etc.) requires modification of their surface for a stronger anchoring of the precursor. Carbon materials functionalized with oxygen or nitrogen-containing groups are more preferable in this view.

#### 3.2.2. Polymers, SiO_2_ and Other Hard Templates

Nanocasting on a hard mesoporous template was applied in several works to produce the structured ceria. For this purpose ordered silica materials such as SBA-15 [27,39] or MCM-48 [40] were used. This method allowed producing highly ordered ceria materials; thus, S_BET_ of MCM-48 templated ceria (225 m^2^/g) is much higher than of its CTAB-templated counterpart. SBA-15 was not so efficient in providing high porosity, S_BET_ of CeO_2_ produced using SBA-15 is up to 120 m^2^/g [39]. The template to precursor (cerium nitrate) ratio, stirring duration, the temperature of the solvent (ethanol) evaporation, and the number of impregnation cycles were varied in [40], but the calcination temperature was the same (550 °C). MCM-48 template was removed by alkali treatment. The best sample with the largest peak of Ce^4+^ reduction on TPR-H_2_ profile referring to the highest extent of cerium reduction was prepared using 50 wt.% of Ce, 30 min stirring duration, high ethanol evaporation temperature and one filling cycle. Note, that similar higher reducibility and defectiveness promising for catalytic and photocatalytic applications were found for small (3-6 nm) ceria particles confined in SBA-15 pores, in contrast to large particles on the surface of this material [82]. SBA-15-templated ceria was tested in benzene oxidation [27], and as a support for Au particles in the catalyst for the aerobic oxidation of 5-hydroxymethylfurfural to 2,5-furandicarboxylic acid important in biomass processing [39]. However, in the last case too large Au nanoparticles were produced to fit pores in ceria, which did not allow to gain maximum advantage from the ordered porous structure. Careful tuning of silica template removal allowed to improve the selectivity to the desired product 2,5-furandicarboxylic acid from 51 to 92%.

Ni-Ce binary oxide systems with different Ni:Ce molar ratios (from 0.43 to 1.66) were prepared using SBA-15 as a hard template in [41]. The TEM micrographs showed both the regions in which the ordered rod-like morphology of the SBA-15 template was replicated and other regions with the collapsed structure. The S_BET_ values of the ceria-based systems (>120 m^2^/g) were higher than those reported in the literature for similar samples. All the Ni-CeO_2_ systems were found to be highly active and selective in the CO methanation reaction after mild reduction pretreatment (H_2_ at 400 °C for 1 h). We can conclude that optimizing the type of template can improve the texture parameters of oxide systems without deteriorating the catalytic properties.

NO_2_-assisted soot combustion over three dimensionally ordered macroporous (3DOM) ceria-based catalysts was studied in [83]. Both the use of the colloidal crystal template of monodisperse spheres of PMMA and the addition of copper via impregnation step contributed to the high efficiency of the catalyst. Authors suggested that copper modification promotes oxygen exchange between ceria and O_2_, thereby improving the concentration of active oxygen. However, microporous structure of templated 3DOM structure deteriorated diffusion of the copper-containing impregnation solution, thus worsened the contact between ceria and copper compared to the reference sample. The advantage of CeO_2_-3DOM was demonstrated in NO_2_-assisted soot combustion, as the templated sample utilized NO_2_ more efficiently than the reference CeO_2_. Part of NO_2_ contributed to the active oxygen production, while macroporous 3DOM structure promotes the transfer of active oxygen species to soot.

The same template and nitrates/citrates of Ce and Pr were used in work [84] as precursors for the preparation of Pr-doped ceria Ce_0.9_Pr_0.1_O_2_. The catalytic activity in CO oxidation of the template-based mixed oxides prepared from nitrates was found to be higher than that of their counterparts synthesized from citrates. This difference can be caused by the higher specific surface area of mixed oxides synthesized from nitrates (118 vs. 51 m^2^/g). Melting and decomposition of the nitrate precursors during the template combustion, followed by partial collapse of the structure accompanied by the appearance of micro and mesopores provided higher concentration of oxygen vacancies in the final mixed oxide catalyst. On the other hand, the presence of meso- and macroporous structure with the well-defined pore sizes was observed for the sample prepared from citrates. The infiltration of nitrates into the PMMA colloidal solution had a stronger positive effect on the catalytic activity in CO oxidation than the formation of an ordered 3DOM structure while using a citrate. Thus, in the case of the hard templates with rigid structure, the type of precursor compound used for the oxide formation plays an important role and can affect the properties of the prepared oxide material. As it was found by XPS, the surface of 3DOM Ce-Pr-O_x_ catalyst can be more easily reduced and re-oxidized than that of the reference material prepared without template, mainly due to the presence of praseodymium cations rather than cerium ones [85]. By the way, PrO_x_ with 3DOM structure is even more active in PROX than corresponding CeO_x_ due to better reducibility [86].

To improve heat resistance, in situ created NiO was used as an unusual inorganic template to produce mesoporous samaria-doped ceria (SDC) that can be used in solid-oxide fuel cells (SOFC) [87]. The precursors of all oxides were mixed in water and chelated with EDTA and citric acid, with SDC particles surrounding NiO grains; then gelation was performed at 90 °C, followed by stepwise thermal treatments at 250 °C and then at 350–900 °C. Afterwards NiO was removed by diluted nitric acid treatment to produce SDC with the favorable thermal stability, narrow pore size distribution and acceptable values of specific surface area that increased from 37 to 85 m^2^/g with the decrease in SDC:NiO ratio from 1:0.5 to 1:9 for the systems calcined at 700 °C. It was found that SDC significantly more suppressed sintering of NiO particles than vice versa. Therefore, this method can be also promising for the synthesis of triple NiO-SDC oxide systems considering the significant mesoporosity of the sample with the narrow pore size distribution (centered at about 20 nm), which is evident from N_2_ adsorption data presented in the article.

Two aspects of the hard-template methods are very important: (i) the technique of template impregnation with the ceria precursor solution, providing the appropriate wetting extent, and (ii) the resistance of the oxide systems to harsh thermal, acid/alkali and other effects during complete template removal.

### 3.3. Combined Methods

Various methods are combined in attempts to produce ceria-based oxide systems. Examples of combining template methods with the non-template ones, hard and soft templates can be found in literature. In [64] a hard template (carbon spheres) was added during solution combustion synthesis to synthesize MnO_x_-CeO_2_ catalyst for CO oxidation. The MnO_x_-CeO_2_ sample prepared by the carbon-assisted combustion method exhibited the best texture and catalytic properties (total CO oxidation was achieved during 5 cycles at 160 °C) compared to its counterparts synthesized by the traditional combustion method without template, and traditional hard template method without organic fuel addition. The article compares the catalyst with the samples of similar composition but prepared using the other hard template (silica microspheres) removed by dissolution in NaOH aqueous solution. It was demonstrated, that carbon spheres not only leave voids in the ceramics after template removal, but also contribute during combustion to a change in the crystallinity of manganese oxides, limit the growth of their crystallites, and increase the content of more oxidized forms of manganese. However, S_BET_ of silica-templated sample was twice higher (115 vs. 54 m^2^/g). The reduction peaks in the TPR profile significantly shifted to lower temperatures for the system, prepared with carbon template, indicating the improved oxygen mobility and reducibility of manganese oxides caused by their strong interaction with CeO_2_. Thus, this complex preparation method keeps the advantages of the traditional combustion synthesis (simplicity and productivity), while improves the catalytic activity due to the changes in microstructure and/or active species oxidation state as a result of the application of carbon spheres as a hard template.

The combination of hard (PMMA) and soft (Pluronic F127) templates for the synthesis of CeO_2_ intended for use as a catalyst in NO_2_-assisted soot oxidation is considered in detail in [65]. 3DOM-CeO_2_ sample prepared by the hard-template method without Pluronic F127 comprises mesopores of around 32 nm. The addition of soft template produces material with small mesopores of 3.5 nm in size. The infiltration of the cerium precursor together with Pluronic F127 into the PMMA template allows forming fine interstitial pores in the body of the hard template (Figure 5).

The porous structure of the synthesized oxide depends on the Pluronic F127 concentration. The ceria sample synthesized with 7.9 mM concentration of Pluronic F127 (Figure 5) demonstrates the best catalytic properties. Thus, combining different types of templates allows fine tuning of the porosity of CeO_2_. This is an attractive way to design noble metal free catalytic materials with the high area of soot-catalyst contacts. According to the “key-lock” concept, these contacts improve transfer of active oxygen from the catalyst to soot particles and, consequently, ensure excellent catalytic activity in oxidation processes.

There are also methods for preparing oxide systems in which ceramics precursor also serves as a template. These processes can be classified as “self-templating” methods. For example, synthesis of porous/hollow structured ceria using partial thermal decomposition of Ce-MOF was reported in [42]. In contrast to the usual formation of voids during complete thermal decomposition of a hard template, in the method used in [42] cerium oxide partially inherits the porous structure of the initial MOF, and the additional internal pore space is formed after removal of the residual Ce-MOF by lactic acid. The gradual increase in the specific surface area after thermal treatment and selective etching of pristine Ce-MOF confirms the formation of porous and hollow structure in ceria. The authors of the work [42] argue that such porous/hollow structure is highly desirable for supports of heterogeneous catalysts because it can facilitate the dispersion of secondary species and enable substrate molecules to easily contact the active sites. However, no catalytic tests were provided.

Nanoparticles of another MOF structure ZIF-8 with nitrogen-rich zeolitic imidazolate framework were used as a sacrificial template to provide a facile method for the synthesis of CeO_2_ nanoparticles under mild conditions [43]. MOF structure provides the in-box nanocage for CeO_2_ formation, and alkali surface of template avoids using external alkaline additives or stabilizers. Etching of MOF template proceeds under the influence of protons generated during hydrolysis of Ce^3+^ ions, which may be partially oxidized to Ce^4+^ by NO_3_^−^ ions. Alkaline conditions resulting from the hydrolysis of 2-methylimidazole from ZIF-8 structure transform both types of ceria ions to Ce(OH)_3_/Ce(OH)_4_, which is dehydrated to CeO_2_ under the influence of NaOH.

The perspectives of the dual template strategy were underlined in [37] reporting the synthesis of another interesting structure—hollow ball-in-tube (HBT) asymmetrical structured ceria using hard (SiO_2_ spheres) and sacrificial template (Ce(OH)CO_3_ nanorods). After templating the composite was washed firstly with NaOH and then with the acid solution. The unique structure of ceria is provided by the different solubility of two templates in alkali solution, as dissolution of SiO_2_ spheres leaves pores in the bulk of the sacrificial template. Ce^3+^ ions released at slow dissociation of Ce(OH)CO_3_ react with OH^−^ to form ceria precipitating on the surface of nanorods and partly dissolved SiO_2_ spheres; in this way the complex structure of the final material is formed. Ceria of HBT morphology is preferable as a support for Au nanoparticles compared to simple hollow CeO_2_ nanotubes. The catalytic activity of Au/HBT-CeO_2_ in CO oxidation was much better, e.g., at 15 °C the TOF value normalized to Au loading was 2.2 times higher than that of Au/CeO_2_. The specific surface area of HBT-CeO_2_ was only 1.3 times higher than that of CeO_2_ nanotubes (102 vs. 74 m^2^/g), so the authors assumed that the stronger metal-support interaction in Au/HBT-CeO_2_ catalyst was a possible reason for its improved catalytic activity.

Other authors [66] reported a complex preparation method of Ce_x_Zr_1−x_O_2_ solid spheres based on the combination of soft (Pluronic F-127) and hard (polymer-based spherical activated carbon) templates. They investigated the influence of Ce:Zr ratio on the properties of the synthesized materials. The catalysts were tested in dimethyl carbonate (DMC) synthesis via direct conversion of CO_2_. It was found that the increase in the ceria content in the mixed oxides decreased the specific surface area except for the systems with Ce:Zr = 0.4–0.5 that exhibited the highest S_BET_ values (about 120 m^2^/g). This resulted from the formation of the structurally homogeneous solid solution at these Ce:Zr ratios. In addition, the amount of basic and acidic sites in the Ce_0.5_Zr_0.5_O_2_ system was the highest among all the counterparts and this catalyst showed the highest DMC yield.

In a broad sense, the systems synthesized by coating of the surface of a porous solid system with a material of the desired composition, where the template is then not removed, in contrast to the sacrificial templates, can also be attributed to hard-templated materials. In this way the very interesting comparison of the influence of pore system parameters on the catalytic properties of ceria in water gas shift process was performed in [88]. In this work CeO_2_ was supported not only on aluminum foams with different porosity (5 and 40 pores per inch), but also on Al sponge synthesized by replica method—supporting of aluminum on the spheres prepared from bread mill and NaCl that were further removed by dissolution in water. The presence of metal inside catalytic layer strongly mitigates heat-transfer restrictions typical for oxide catalysts. Better performance in water-gas shift reaction was observed for the ceria on the replicated Al-sponge, than on compressed Al foam, due to different distribution of pores. The sponge had a regular structure comprising spheroidal pores with limited interconnection; the porosity of the compressed foam was less regular and anisotropic which cannot be completely eliminated by the compression.

The ceria precursor was confined between walls of 3D mesoporous silica KIT-6 and Pluronic 123 by solid-phase grinding [34]. The size-controlled ceria particles were produced by one-step calcination of this material. The authors underlined the greenness of this synthesis that provided confinement of ceria precursor and its conversion into ceria in a single stage. What is particularly important is that decomposition temperature of the chosen template is higher than that of cerium nitrate used as a ceria precursor, providing excellent dispersion of the produced ceria on the KIT-6 surface. S_BET_ of these systems exceeded 300 m^2^/g, so they showed the high thiophene adsorption capacity (0.14 mmol/g for the material comprising 20 wt.% CeO_2_) and high performance in hydrodesulfurization—high activity, good stability, and recyclability.

Such complex synthesis techniques combining the advantages of various types of templates provide the means to form ceria-based structure with unusual and very promising properties for application in catalysis. However, these techniques are more complicated than the conventional template methods, which significantly affects their reproducibility and cost.

## 4. Bio-Templated CeO_2_-Based Catalysts

Biotemplates form a large and very promising from environmental point of view group of templates. Most natural biomaterials are renewable and can be harvested in large amounts at low costs; this application also allows recycling of secondary biomaterials. Nature provides a huge selection of biomaterials with a variety of textures, from which one can choose suitable for use as a template to produce oxide materials with desired texture and morphology, including ceria-based oxide composites. Moreover, natural biomaterials often exhibit a multi-scale structure, in contrast to synthetic organic compounds, where special efforts must be applied to form hierarchical porosity.

The biotemplate nature strongly influences the structure and porosity of the synthesized oxide. A variety of biomaterials, such as lignocellulosic biomass and products of its processing [58,59,60], wood-based materials [48,49,50,57,89], plants’ leaves and flowers [52,53,58], cotton [90,91], eggshells [55], materials of animal origin [92,93], microorganisms [54,94,95] and so on can be used as biotemplates.

Biomass suitable for use as templates can be divided into plant and non-plant groups (Figure 6), and within these two groups into (i) biological materials which are used without deep processing (e.g., wood sawdust, leaves, husk, peels, animals’ scale, yeast etc.; and (ii) products of biomass processing (e.g., paper, tissue paper, isolated biomass components).

Materials synthesized using biotemplates reproduce well the complex structure of biological objects, as it is shown in Figure 7 for CeZrO_x_ composite produced using pine sawdust as a template and described in [50].

### 4.1. Cellulose and Wood Fiber Templates

The first works published in early 2000s by the researchers from Ford Motor Co. [96,97] and demonstrated the use of cellulose-containing biomass processing products as templating agents played an important role in the development of novel biotemplate methods to produce oxide ceramics. They revealed that the adsorption of oxide’s precursors on the surface of various types of paper allows producing materials that inherit the morphology and texture from the original cellulose-containing template.

Here, some recent examples of using biotemplates in the synthesis of ceria-based systems and their catalytic application are considered.

Hierarchical porous nanocrystalline ceria (nanocrystals diameter of about 6–8 nm) was successfully synthesized using filter paper as a biotemplate [61]. The obtained CeO_2_ replicated the network of cellulose fibers with a diameter of 1–3 µm. The synthesized material showed a relatively high specific surface area of 70 m^2^/g and narrow pore diameter distribution in a range of 2 to 4 nm. The comparison of catalytic activity in degradation of acid fuchsin revealed that the hierarchical porous CeO_2_ catalysts were the most active compared to commonly used CeO_2_ and non-catalytic reaction. The high catalytic efficiency can be explained by the higher specific surface area and increased content of active surface oxygen, which was confirmed by the TPR method. The properties of the prepared biomimetic sample were promising for the possible catalytic applications, especially in the field of wastewater treatment.

Fibrous ceria replicating the original template with the particle diameter from 9 to 5 μm, and the length up to 1000 μm resulted from the annealing at 600–800 °C of cellulose impregnated with the ethanol solution of cerium nitrate [98]. The authors underlined that the completeness of template removal can be confirmed by the absence of carbon in energy dispersive spectra and C–O, C–C and C–H lines in the IR spectra. The catalyst annealed at 800 °C was more efficient in the methyl orange photo destruction (pH 3.5, acetate buffer solution) and UV phenol photooxidation than the one annealed at 600 °C with lower crystallinity and particle size, comprising carbon moieties from incompletely removed cellulose, and the counterparts prepared without cellulose template.

A hierarchically structured porous CeO_2_ catalyst was also prepared using nanocellulose that is a renewable plant-derived template nanomaterial [58]. 3D-network of the porous structure can be varied by modifying the content of cellulose nanofibrils, nanocrystals, and alginate in the templating suspension. The nanocellulose introduced into suspension imparted a controlled, well-defined porosity to the CeO_2_. The use of cellulose nanocrystals led to the formation of fingerlike species of CeO_2_ catalyst that are larger than those imparted by the alginate alone.

Mesoporous CeO_2_ synthesized using microcrystalline cellulose as a template was active in catalytic ozonation of phenol [59]. Templated CeO_2_ had a moderate specific surface area of 50 m^2^/g that is approximately 5.5 times higher than that of the material prepared without template. The enhanced efficiency of the biomorphic catalyst in phenol removal resulted from both the improved specific surface area and the presence of Ce^3+^/Ce^4+^ redox couple on the surface. Such environmentally important indicator as the chemical oxygen demand decreased to a much greater extent with ozonation in the presence of cellulose templated CeO_2_.

Filter paper which is also composed of cellulose but has a braid structure was used as a template to prepare CeO_2_-TiO_2_ composites for CO oxidation [60]. The titania-ceria composites reproduced the morphology of the template and were braided from fibers having diameters in a range of 1–6 µm and lengths of several hundred micrometers. The synthesized composites showed higher surface oxygen activities at low temperature and larger specific surface areas than those of pure ceria. The sample with the Ce:Ti mole ratio of 8:2 and S_BET_ of about 80 m^2^/g showed a high catalytic activity in CO oxidation (50% CO conversion at 280 °C) due to the presence of CeO_2_-TiO_2_ solid solution, whereas the excessive TiO_2_ doping resulted in the mixed phases formation deteriorating the catalytic properties.

Cellulosic fibrils from the micro fiber bundles of banana pseudo stem were used as a biomorphic template for synthesis of ZnO/CeO_2_ composite nanowires [99]. The produced material showed a smooth, spherical morphology. The XRD confirmed the formation of highly dispersed crystalline material with different ZnO:CeO_2_ ratios. The catalyst was successfully applied for the photodegradation of Direct Red dye under UV and direct solar light irradiation.

ZnO-CeO_2_ composites with the hollow fiber morphology were synthesized using cotton as a biotemplate [90]. Ceramic fibers were long, continuous, and randomly oriented with porous walls of about 2 μm thick (Figure 8a). The outer diameter of fibers was in the range of 8–14 μm. The physicochemical analysis revealed the polycrystalline biomorphic porous structure of ZnO-CeO_2_. The synthesized material was used as a gas sensor. ZnO-CeO_2_ showed the highest response to ethanol at 260 °C with better stability and selectivity compared to pure ZnO.

Another fibrous biomaterial used for ceria synthesis is natural kapok fiber. The produced CeO_2_ well remained the tubular morphology of kapok template (Figure 8b) [100]. Gas sensors for H_2_O_2_ and glucose detection were fabricated on the base of hierarchically porous hybrid composite comprising CoFe-LDH (LDH—layered double hydroxide) and biomorphic CeO_2_. CoFe-LDH nanoflakes covered perpendicularly or aslant the surface of CeO_2_ forming the hierarchical surface morphology (Figure 8c). Particles of this material were resistant to agglomeration. The authors declared that such sensitive platform is very promising for the detection of H_2_O_2_ and glucose in practical concentrations due to its acceptable sensitivity, good selectivity, and easy of separation.

CeO_2_, Co_3_O_4_ and mixed Co_3_O_4_-CeO_2_ hollow microfibers for soot oxidation were synthesized by incipient wetness impregnation using cotton as biotemplate [91] with subsequent calcination at 600 °C. The fibers ca. 10 μm in diameter are formed by bundles of smaller fibers and are hollow inside. Modification with cobalt oxide did not affect the Ce^3+^/Ce^4+^ surface atomic ratio. The ratio of lattice oxygen to labile surface oxygen species determined by XPS decreased in the series Co_3_O_4_ > Co_3_O_4_-CeO_2_ > CeO_2_ and correlated with the catalytic activity in soot oxidation. Interesting, that the variation of Co_3_O_4_ loading (2 and 12%) and the method of its addition (one-pot or post-impregnation) had almost no effect on the catalytic properties, and this result requires more careful study.

### 4.2. Raw Biotemplates

A special class of templates is renewable biological materials that can be used without pre-treatment. These templates include various types of plant species (wood, grass, sawdust) and biotemplates of animal origin (e.g., fish scales, eggshells).

#### 4.2.1. Plant Biotemplates

The waste materials such as wood sawdust attract special attention as templates because they comply with the main principles of green chemistry. Moreover, these materials usually have a regular pore structure.

Eastern white pine replicated CeO_2_ [48] was synthesized by the Pechini method. Despite the high calcination temperatures (1200–1500 °C), the wood structure was well preserved in the synthesized oxide. The produced catalysts were active and stable in the thermochemical production of CO from CO_2_ in a cyclic manner.

The use of bark layer of some trees is especially attractive because of the fast regeneration: the bark of these trees can be harvested each 9–13 years, since during this time it is completely restored [49,57]. For Mediterranean evergreen oak tree, the regeneration time is much shorter. Biomimetic cork-templated CeO_2_ eco-ceramics was proposed for hydrogen generation using concentrated solar energy [57]. The produced ceramic material had a 3D-ordered macroporous cellular structure, replicating the extremely porous cork morphology with elongated hexagonal cells of about 20 μm diameter, of 40–50 μm in length, and very thin walls of around 1 μm. The advantages of the cork templated CeO_2_ is the presence of channels crossing walls, which allows gases to permeate into the structure, greatly increasing the accessibility of active sites. Ceria was also synthesized using cork-template granules and compared with the counterparts templated by polyurethane foam [49]. The activity of the produced materials was measured in solar thermochemical CO_2_-splitting cycles. The CO yield over cork-templated ceria was about twice higher than that over ceria foam. Cork-derived CeO_2_ also showed two times larger reaction rates and improved stability, demonstrated in 11 reaction cycles. The advantage of cork-derived CeO_2_ is that the mean cell size is at least one order of magnitude smaller than that in the polymer-derived ceria.

Wood continues to be a popular template, and a variety of tree species make it possible to regulate the morphology of the resulting ceramics. Sr-doped ceria ceramic was prepared by the wet impregnation of Linden wood [89]. The wood structure was well replicated by the ceramic material while maintaining tracheidal pore channels and pits. The synthesized Ce_0.9_Sr_0.1_O_2_ had porous channels with the diameter of about 50 μm. The walls of the ceramic material were micro-mesoporous with the pore radius between 2 and 6 nm. It is difficult to judge the catalytic activity of this material because the authors suggested its application for the radioactive ^90^Sr isotope storage. The Raman spectroscopy study showed the presence of not only intrinsic oxygen vacancies but also the additional O^2−^ vacancies that were introduced into the ceria lattice by the substitution of Ce^4+^ with Sr^2+^ ions. Such materials can be promising for catalytic applications.

Chip and readily available wood sawdust, which is the industrial waste, can be effectively used instead of organic templates for the preparation ceria-based catalysts of important catalytic processes. Thus, pine sawdust was used for obtaining CeZrO_x_ mixed oxides for CO oxidation [50]. Ce-Zr oxide samples prepared by wet impregnation of pine sawdust with water solution of ceria and zirconia precursors followed by calcination were tested in total CO oxidation at 100–400 °C and compared with the similar systems synthesized using the CTAB template by the EISA method. The decrease in calcination temperature from 600 to 500 °C enhanced the S_BET_ value. The biomorphic Ce-Zr oxide system demonstrated a porous structure similar to that of the original sawdust (Figure 7) and much higher catalytic efficiency than the CTAB-templated sample, despite two times lower S_BET_. Earlier in the literature, we did not see any mention of the preparation of a cerium-zirconium catalyst using a biotemplate, which is more efficient in the oxidation of CO than an analogue obtained using CTAB as an artificial template. According to the EDS and XPS results, biomorphic CZ inherited from the biotemplate not only the morphology, but also the ash impurities, such as Ca and K. Intercalation of the ions of alkali and alkaline earth metals into Ce-Zr oxide lattice can contribute to the increase of the active oxygen fraction in the biomorphic samples compared to the CTAB-templated ones. However, other authors [48] believed that the content of specific metal ions inherited from the wood after the calcination step was too low to affect the activity or thermal stability of the ceramic materials. They found that ceria inherited from biotemplate only trace amounts of ash impurities except for Mg^2+^ (0.016 mol/mol), K^+^ (0.11 mol/mol), and Ca^2+^ (0.004 mol/mol) (all values are presented per one mol of Ce). According to this work, not dopants but the unique pore structure and the uniform distribution of active sites on the surface of oxide system were the major factors that improved the catalytic activity. On the contrary, in [101] the positive influence of Ca and K addition on the catalytic activity of CZ materials in CO oxidation was directly demonstrated by the addition of these dopants during precipitation of double oxide.

Interestingly, modification with Cu improved the low-temperature efficiency of CeZrO_x_ systems, prepared using both CTAB and wood sawdust as templates; however, the catalytic properties of Cu-modified biomorphic ceramic were inferior to those obtained with CTAB [50].

The Loofa sponge derived from the fruits of tropical liana was used as a biotemplate to prepare composite CuO/CeO_2_-ZrO_2_ catalysts with tubular porous structure [102]. Multi-component structure together with unique morphology provided high activity of this catalyst in soot oxidation, which was studied at temperature-programmed oxidation (TPO) conditions at loose and tight contact between soot particles and the catalyst. This reaction was facilitated by the porous structure of the material that included many brittle multi-channels and fractured sheets, which ensured tight contact between soot particles and the catalyst surface. The redox equilibrium Cu^2+^ + Ce^3+^ ⇆ Cu^+^ + Ce^4+^ on the surface and in the crystal lattice of the catalysts provided an increase in the number of surface oxygen species. The extra copper ions entered the cubic lattice of CZ led to the phase separation to form CeO_2_-ZrO_2_ system, which negatively affected the catalyst performance.

Some biotemplates, such as lotus pollen and yeast, provides the microspheric morphology. Lotus pollen was used for the synthesis of CeO_2_ microspheres of 10–15 μm in diameter [52]. The walls of the spheres have a pronounced mesoporousity. The lotus pollen not only acted as a template, but it also doped the synthesized ceramics with nitrogen. The N-doped CeO_2_ spheres showed a higher photocatalytic activity compared to pure bulk CeO_2_ and undoped CeO_2_ spheres. XPS study revealed the enhanced surface oxygen activity for the biomorphic ceria spheres.

The pollen-templated Co_3_O_4_/CeO_2_ composites comprised hollow microspheres with the external diameter of about 26 μm [53]. The specific surface area of this material was moderate (48 m^2^/g). The biomimetic Co_3_O_4_/CeO_2_ catalysts were found to be more active and stable in photocatalytic degradation of methylene blue (MB) and tetracycline (TC) than their co-precipitated counterpart.

Agricultural wastes, such as stems of common plants, were also proposed as biotemplates. Several oxide materials were produced using long rape flower stem as a template. Since rapeseed is grown in large quantities for oil and biodiesel, the straw of this crop is a widely available waste for processing. Thus, in this way biomimetic material comprising cerium oxide nanosquares on reduced graphene oxide (RGO) was prepared [61]. The peculiarity of this synthesis was the incomplete removal of the template. The biomaterial impregnated with cerium nitrate was converted into graphene oxide during high-temperature pyrolysis in an inert atmosphere. RGO nanosheets replicated the honeycomb-like structure of the rape flower stem. RGO served as a stabilizer for tiny CeO_2_ particles which were anchored on graphene sheets. The synthesized biomorphic composite, proposed as photocatalyst for solar energy conversion, had a large specific surface area of 359 m^2^/g, high electrical conductivity, micro-meso porous structure, intimate contact between RGO and CeO_2_, and a narrow band gap due to the presence of abundant oxygen vacancies, arising due to high concentration of surface defects in RGO and tiny sizes of ceria quantum dots. All these features favored the process of photocatalytic water splitting: H_2_ production over biomimetic sample was four times higher than over its counterpart prepared without template (about 800 and 200 μmol H_2_/g, respectively).

The same template and synthesis technique were used for preparations of multi-component composites Co_3_O_4_/CeO_2_ on graphene [103]. The honeycomb-like original template structure is replicated by 3D graphene structure with highly ordered macropores of about 35 μm in diameter. The surface of 3D graphene sheets was completely covered with Co_3_O_4_-CeO_2_ nano-particles. Such structure provided plenty of conducting channels for electron transfer between the electrolytes and electrode.

Biomorphic porous CeO_2_ powder, retaining the porous structure of the original biotemplate, was also synthesized by the hydrothermal biomineralization method using stems of clover [51]. Several tricks were used to improve the properties of the resulting ceria: (i) pretreatment of the stems by ethanol and HCl solutions provided adsorption centers (–COOH and –OH groups) to facilitate anchoring of the precursor on the surface of the biomaterial by biologically induced mineralization; (ii) the addition of hexamethylenetetramine to the reaction mixture ensured the presence of CO_3_^2−^ ions arising from its hydrolysis; these ions interacted with Ce(OH)_2_ to form the solid CeCO_3_OH precursor under supersaturation conditions; (iii) hydrothermal conditions under which the desired reactions proceeded in a simple, efficient and eco-friendly manner. The material contained macropores with sizes from several to dozens of micrometers and mesopores with the bimodal pore size distribution centered at 15 and 35 nm, according to N_2_ physisorption data. This mesoporous structure resulted from the cell walls of the stems. The authors evaluated the possible applications of synthesized CeO_2_ in oxygen sensors and three-way catalysts. The Oxygen Storage Capacity (OSC) value, obtained from the TG analysis, was more than twice higher for the biomorphic CeO_2_ than for the comparative powdered CeO_2_. The delocalized oxygen vacancies, weakly bonded oxygen and interstitial oxygen ions in the defective crystal structures were suggested as the reasons for the difference in OSC. Biomorphic CeO_2_ showed the improved catalytic performance in catalytic decolorization of acid magenta solutions.

Diatom algae surrounded by a silicon dioxide cell wall (frustule) is an extremely attractive biotemplate material. Its structure is in many ways similar to that of artificial 3DOM silicon dioxide. This biotemplate was effectively used for the synthesis of hierarchically porous nanostructured CeO_2_-based systems from nitrate precursor [54]. Surface mesopores of about 5 nm in diameter were covered with fine CeO_2_ crystallites of approximately 8 nm in size. According to the H_2_-TPR, the biomorphic ceria was more easily reducible, as the peak of its reduction was shifted to lower temperature for 150 °C and comprised higher amount of surface oxygen. XPS confirmed this observation showing higher concentrations of Ce^3+^ and more oxygen vacancies in biomorphic CeO_2_. As a result, its catalytic activity in CO oxidation was greatly improved compared to bulk CeO_2_ in the whole tested temperature region (150–400 °C). The onset temperature of CO oxidation and the temperature of 50% CO conversion for bio-templated sample were lower by 55 and 65 °C, respectively. The hollow internal space of biomorphic hierarchical mesoporous CeO_2_ was considered by the authors as the additional advantage that facilitates the transport of molecules to active sites during CO oxidation. However, the completeness of the removal of the template during calcination at 500 °C was not discussed in the work.

#### 4.2.2. Biotemplates of Animal and Microbiological Origins

Small CeO_2_ mesoporous hollow microspheres (1.5–2 μm in diameter, S_BET_ = 39 m^2^/g) were prepared using the yeast template [94]. The authors suggested that the yeast acted as a solid frame for the deposition of cerium hydroxide. The TPR-H_2_ profile revealed the hydrogen consumption peak at 520 °C corresponded to the reduction of surface oxygen. The intensity of this peak was significantly higher than that for commercial CeO_2_. The morphology with hollow internal space improved the mass transport of CO and oxygen molecules to the active sites during CO oxidation, providing the enhanced activity of the yeast templated CeO_2_ in CO oxidation compared to the commercial one.

Yeast was also used as a template for the synthesis of CeO_2_ and Fe-doped cerium oxide hollow microspheres for visible light photodegradation of acid orange 7 (AO7) [95]. The Fe-doped CeO_2_ hollow microspheres showed a relatively higher proportion of Ce^3+^ and concentration of O_2_^2−^ because of the Fe^3+^ ions incorporation into the crystal lattice of CeO_2_. The incorporated Ce^3+^ and oxygen vacancies are necessary to improve the charge compensation. As a result, Fe-doped CeO_2_ hollow microspheres exhibited a higher photocatalytic performance in degrading AO7 aqueous solutions containing H_2_O_2_ under visible irradiation.

Eggshell membrane was used as a natural biotemplate of non-plant origin to produce ceria [55]. As usual, biomorphic ceria replicated the fibrous nano-porous structure of protein eggshell membrane after calcination at 600 and 800 °C, but the increase in temperature above 1000 °C led to the disappearance of the biomorphic morphology, and spherical ceria grains were formed. Since the specific surface of all samples is below 5 m^2^/g, the method proposed in [55] is not too attractive for the catalyst preparation. The authors suggest the application of the material for radioactive isotopes storage. 

Eggshell membrane is not the only part of the egg suitable for templating. In [93] fresh egg white was used as an eco-friendly foamy template for obtaining size-controlled ceria nano powders. Cerium cations (Ce^3+^) were involved in the electrostatic complexation with oppositely charged proteins molecules from egg white. The authors of this publication suggested that proteins with a large number of −OH groups covered the different faces of CeO_2_, which led to the controllable crystal growth and formation of small, stable, and phase-pure crystalline CeO_2_ nanoparticles. Unfortunately, no characteristics of pore structure were presented in this work aimed at cytotoxicity study.

Fish waste plays a similar role among animal templates as sawdust among plant templates, because the former is a common and cheap material required recycling. Crucian fish scales were successfully used for the preparation of biomorphic nanocrystalline ceria [92] with the high specific surface area of 114 m^2^/g. Its biomimetic structure consisted of 80–100 nm thick sheets that formed pores of 2–10 nm in size. According to the TPR-H_2_ data the synthesized biomorphic ceria showed enhanced surface oxygen mobility, improved reducibility of bulk oxygen (reduction temperature was about 600 °C in contrast to 766 °C for non-templated CeO_2_), and improved catalytic activity in CO oxidation, providing higher CO conversion in the whole studied temperature range (50–500 °C) compared to non-templated counterpart. Even at 500 °C the maximum CO conversion over CeO_2_ prepared without a template was only 80%. In the case of bio-CeO_2_ the CO conversion exceeded 90% already at 400 °C. The advantages of the fish scale templated ceria sample were attributed to the unique sheet structure, high specific surface area and surface oxygen vacancies generated by the reduction of Ce^4+^ to Ce^3+^.

The above presented results clearly show that the type of template affects the reduction temperature of bulk CeO_2_. Indeed, the comparison of the TPR-H_2_ profiles presented in [61], where ceria was synthesized using filter paper cellulose sheets, with that from the reference [92] shows that the peak corresponding to surface oxygen reduction is situated at about 520 °C for all the samples. The position of the high-temperature peak that corresponds to bulk CeO_2_ reduction only slightly changes for filter paper templated CeO_2_ relative to non-templated sample (740 and 770 °C, respectively), but this peak shifted to 600 °C in the case of fish scale templated material. It is difficult to say how exactly this difference affected the catalytic activity, because two samples prepared using animal and plant templates were tested in different reactions, but the effect should be significant.

A green technique has been developed in [104] to prepare hierarchical biomorphic ZrO_2_-CeO_2_ using silkworm silk as a template. After template removal by calcination the sample demonstrated a complex structure which was different for outer and inner part of walls: the outer part was denser with a thickness of about 2 μm, the inner part was more porous, sponge-like. In the produced ZrO_2_-CeO_2_ mesoporous material the fibers were assembled into crystallites of 10–20 nm in size. The authors considered synthesized materials as promising for catalysis because of the hierarchical porosity and “high” specific surface area of about 12 m^2^/g. However, this value can hardly be treated as high in view of catalytic application.

#### 4.2.3. Biopolymers, Extracts and Amino Acids as Biotemplates

The interesting litchi-peel-like hollow CuO/CeO_2_ structure in which secondary hemispherical hollow shells were attached to the main hollow microsphere of several micrometers in size was synthesized by the one-step aerosol spray pyrolysis using dextrin as a sacrificial template [105]. The properties of the structure could be tuned by varying the amount of dextrin in the sprayed solution, namely, litchi-peel-like structure was produced only at relatively high dextrin concentration. 20 wt.% CuO/CeO_2_ catalyst with this morphology and S_BET_ of about 50 m^2^/g was superior in activity in CO oxidation not only to the hollow catalyst with a conventional structure, but also to the noble metal 5%Pd/Al_2_O_3_ catalyst (T_50_ was 83, 109 and about 130 °C, respectively). It was also more stable during 40 h time-on-stream both in the absence and in the presence of water in the reaction mixture. The excellent catalytic properties were explained not only by the typical reasons (improved specific surface area, redox properties, and CuO dispersion), but also by the presence of step-promoted stable interfacial active sites.

Hierarchically porous ceria as a support for Au/CeO_2_ catalyst was prepared by the citrate sol-gel method using a bovine serum albumin (BSA) scaffold [56]. The BSA template consisted of multiple acid residues and showed strong binding ability that help to encapsulate and stabilize Ce^3+^ from the precursor. In addition, the complex secondary structure of BSA comprising α-helix, β-sheet, β-turn, and random coil led after calcinations to a hierarchical porous structure of ceria with plenty of oxygen vacancies. The increase of BSA amount from 0 to 0.7 g during synthesis enhanced the catalyst performance, because the template retarded the sintering of ceria crystals during thermal treatments. The test in benzene oxidation, modelling the VOC disposal, demonstrated the high catalyst activity (90% benzene conversion at relatively low temperature of 210 °C, stability during 140 h time-on-stream) due to the presence of Au nanoparticles of 3.2 nm in size and high concentration of mobile oxygen on the surface of the BSA-CeO_2_ support.

Amino acids were used as templates of animal origin in several works. Porous 3D CeO_2_ structures were synthesized with Glycine amino acid [63], the use of which can be classified as a soft-template method. The amount of Glycine affected the shape of the synthesized ceria. The increase in Glycine concentration resulted in the disappearance of spherical particles, and formation of bowknot structures. Varying the amounts of precipitation agent and Glycine the authors determined the optimal ratio at which the highest concentration of oxygen vacancies and relatively narrow energy gap was observed in the produced ceria. These characteristics are important for catalytic applications. The temperature of the complete CO conversion over the best catalyst was by 250 °C lower than that for bulk CeO_2_. The nature of amino acid did not influence the morphology of CeO_2_. Thus, the use of l-Glycine, l-Lysine, and l-Proline as templates led to similar bowknot or coral-like structures with a narrow center connecting two cones and wider diameters at the sides [62]. Ceria formed compacted linear nanotubes with the average diameter of about 200 nm. However, the XRD results revealed different sizes of CeO_2_ crystallites depending on the amino acid nature. The mesostructured CeO_2_ samples templated with l-Lysine and l-Glycine contained crystals of about 15 nm in size, while those prepared with l-Glutamic acid, l-Aspartic acid, and l-Valine were formed by 12 nm crystals, and finally those prepared with l-Proline and l-Histidine contained the smallest crystals of about 6 nm. Unfortunately, the authors of work [62] described only the physicochemical characteristics of the synthesized oxides and presented no data about their use, neither in catalysis nor in the other fields.

#### 4.2.4. Biotemplates for the Synthesis of Ceria Nanoparticles

It is also important to separate a group of template methods in which biomaterials are used for the synthesis of nanoparticles. The formation mechanism of such particles involves the intra-atomic interaction between cerium ions from the precursor with functional groups on the surface of a template, such as –COOH, –OH, and –NH_2_. For this purpose, chitin, the commercially available nontoxic and renewable natural material, and its deacetylated derivative—chitosan comprising plenty of reactive amino side groups are especially effective [106].

For example, chitosan powder [107,108], and polysaccharide polymer including pullulan [109] were used as natural matrices for CeO_2_ nanoparticles synthesis. The structural oxygen atoms in the chitosan precursor coordinated with solvated Ce^3+^ ions [107]. During hydrolysis, the Ce^3+^ was converted to Ce^4+^ forming highly crystalline CeO_2_ nanoparticles stabilized by chitosan [108].

Chitosan was also used for the synthesis of ceria nanoparticles of 4 nm in size with the relatively high S_BET_ of 105 m^2^/g [110]. The final material was produced by the calcination of chitosan-ceria hybrid spheres. The interactions between Ce ions and chitosan were confirmed by FTIR analysis. The absorption spectrum of CeO_2_ nanoparticles indicated a direct band gap of 4.5 eV that opens the way for its photocatalysis application.

Chitosan-templated CeO_2_ nanoparticles are active in photocatalytic degradation of Congo Red as a model aqueous pollutant [111]. Since chitosan template is removed by calcination, the treatment temperature can influence morphology and properties of the produced material. Ceria calcined at higher temperatures demonstrated the improved photocatalytic degradation activity because of the larger particle size and increased crystallinity.

Vegetable oils and saponin were used as a part of the templating emulsion to produce ceria for medical application [112]. The mixture of lemon and corn oil with Tween-80 and saponin was emulsified in water and the droplets of this emulsion were used as a template, which was further removed by lyophilization. Ceria nanoparticles produced in this way had the average size of 4.5 nm, S_BET_ of about 55 m^2^/g, and comprised mesopores of 2–10 nm in size, which are good characteristics for catalytic use. However, the unusual N_2_ adsorption-desorption isotherms could result from the incomplete template removal.

The effective synthesis of nanosized ceria was performed using a combination of cerium salt and plant extracts [113,114]. The CeO_2_ nanoparticles were successfully synthesized in the presence of *Eucalyptus globulus* leaf extract by the hydrothermal technique [113]. During synthesis step a network complex structure was formed from hydroxyl groups of 9, 12-octadecatrienoic acid chains from E. globulus extract and Ce^4+^. Further calcination led to a slow decomposition of the polymeric network chains to form CeO_2_. The preliminary results revealed that the synthesized CeO_2_ nanoparticles are promising as a photocatalyst and a cytotoxic agent against human cancer cell lines.

*Linum usitatissimum* L. seeds extract was used for synthesis of CeO_2_ nanoparticles for biological applications [114].

Summarizing this section, biotemplating provides not only a wide range of morphologies and different pore structures of ceria-based ceramics, but in most cases, it also increases the mobility of oxygen in the lattice. Not all biomaterials tested as templates can be classified as widespread and easily available, however, their use is justified by the advantages of the complex hierarchical structure of the synthesized materials. In addition to morphology and porosity inherent to biotemplates, the chemical composition of the template plays an important role in the synthesis of bioceramics. Firstly, the complex and often variable composition of biological materials impairs the reproducibility of the properties of the synthesized ceramics. Secondly, the chemical composition of the template (e.g., the presence of amino groups) often provides good opportunities for anchoring oxide precursors, which, for example, in the case of carbon templates has to be achieved by template modification. Ash impurities remained in the resulting oxide after template removing can serve as catalyst promoters. As a result, a great variety of ceria-based materials very promising for catalytic application were synthesized.

## 5. Conclusions and Perspectives

The following trends can be identified in the field of templated preparation of cerium oxide and related materials. Both biological and artificially prepared materials continue to be widely used as templates for the preparation of cerium oxide and its derivatives. The use of organized carbon structures obtained by the preliminary or in situ pyrolysis of biological materials as templates is expanding.

Artificial templates have a constant composition, and therefore the ceramic materials obtained on their basis have reproducible properties. However, the pore shape and their distribution in the material are difficult to control, especially in the case of soft templates, and changing the type of template even within one family requires serious and unobvious changes in the synthesis technique. In contrast, replacing one template of plant origin with another does not require a change in the synthesis procedure, but allows one to vary the size and shape of pores in the final ceramics within a wide range.

The most popular artificial templates are still CTAB and polymers from the Pluronic family. When used for pure ceria preparation they certainly help improving the catalytic properties in various oxidation reactions, providing enhanced oxygen mobility, developing mesoporous structure and appropriate morphology. However, as it was found in our work [31], when modified ceria materials are synthesized, the stability of the respective complexes should be carefully evaluated because it can crucially affect the porosity of synthesized oxides; thus, Mn addition in the EISA method using CTAB template deteriorates texture properties.

Hard templates like polymers (PMMA), carbon materials of the biomass and artificial origins, structured SiO_2_ and large variety of biomaterials are very promising for the synthesis of ceria-based catalysts due to their ability to create 3D structures and achieve higher specific surface areas (200 m^2^/g and more) than in the case of the soft template methods. The hierarchical structure of catalyst is particularly important for the applications in which large substrate molecules (e.g., natural polymers and other products of biomass processing) or aggregates (soot particles) are converted. The wide range of available carbon materials makes their application especially attractive. However, the surface of carbon materials often needs functionalization to strengthen the bonding between the precursors of cerium oxide and its modifiers and the surface of the template. It can be achieved via acid treatment with nitric or other acids. This treatment creates N- and O-containing functional groups that can serve as adsorption centers. Several templates of animal origin, e.g., amino acids, chitin etc., already comprise these functionalities. The presence of functional groups of a basic nature on the template surface can facilitate the formation of cerium hydroxide and then oxide without adding a precipitant during the synthesis.

Although the traditional artificial templates, CTAB and Pluronic, retain their dominant place, the current trends in the soft and hard template synthesis shift to the combination of methods to increase the efficiency and achieve the specified characteristics of the synthesized oxide, including improved specific surface values and hierarchical pore structures.

Since the specific surface area and pore size are critical characteristics of catalysts, it is interesting to trace the effect of the type of the chosen template on them. Biological templates of plant origin, as a rule, are characterized by the presence of large pores; therefore, ceramic materials obtained on their basis have a smaller surface and a larger pore size in comparison with those obtained using artificial templates. For this reason, biological templates are advantageously selected for preparing catalysts intended for processing relatively large species, for example, soot particles.

At first glance, the use of biotemplates looks like a simple and convenient method for synthesizing cerium ceramics. Biotemplates are renewable resources, so biotemplating is beneficial and promising from the point of view of current global trends towards the preservation of non-renewable raw materials. A wide variety of morphologies of biomaterials, as well as pore sizes and arrangement, provide tools for adjusting texture properties. However, the dependence of the biomaterial composition on natural conditions makes it difficult to obtain ceramics with reproducible properties. In addition, the complex composition and structure of such templates complicates the processes during synthesis. In several works, in which the authors paid considerable attention to the mechanisms of these processes, it was possible to synthesize cerium oxides with the unusual structure, morphology, and texture, and with very impressive catalytic properties. For example, a few works [37,42,75,94] describe the synthesis of hollow ceria particles and structures with mesoporous walls, exhibiting not only high catalytic activity, but also amazing stability in flow systems. In addition, a detailed study of the mechanisms makes it possible to reveal the role of auxiliary substances—ash impurities, complexing agents, solvents, precipitants, swelling agents, and with their help to further increase the catalytic efficiency of the final oxide systems.

Cerium oxide, both pure and modified by the addition of structural modifiers such as zirconium dioxide, is used not only as a catalyst, but also as a promising support for catalytic systems. In many works [31,32,33], methods of adding a catalytically active modifier are compared. Often, the comparison involves one-pot method, when all the components are added to the initial solution during the template synthesis, and different variants of impregnation method where ceria is obtained by the template method and the catalytically active component is added later in a separate step. Typically, the one-pot method provides better interaction between all components of the catalyst system, in addition to higher S_BET_. During the impregnation step, the total pore volume often decreases, the average pore size increases due to the closure of some micropores. Therefore, the choice between the two techniques can be made taking into account the desired texture and morphological characteristics. In cases where the reaction proceeds at the interface between the phases of the support and the modifier, the post-deposition of the active component will be preferable, as was demonstrated in our work [31].

The catalytic applications of the materials discussed in our article are summarized in Table 1. It is seen that templated ceria-based materials find wide applications in various fields of catalysis. Research is ongoing to develop the improved catalytic systems based on cerium oxide for oxidation of soot, volatile organic compounds, and CO, both in the absence and presence of hydrogen (PROX), and photocatalysis. Cerium oxide materials are most widely used in these four types of processes. To carry out oxidation reactions at low temperatures, the template methods are used in the synthesis of gold, Pt and Pd-modified cerium oxides. However, many recent works are aimed on the development of catalysts containing no noble metals, in which ceria is modified with manganese, copper, praseodymium, cobalt, and other oxides. It is important to note that complexly organized materials based on cerium oxide, which are obtained by combined template methods (3DOM, hollow ball-in-tube, hollow spheres, etc.), have shown exceptionally good catalytic properties in dimethylcarbonate synthesis by different ways, processing of biomass derivatives and other promising areas of catalysis.

In a separate group of publications, the template methods were successfully used not to produce ceria materials with the developed porosity, but to synthesize supported and unsupported nanosized ceria particles. They are mainly intended for medical applications but can also be used in catalysis including electrocatalysis. However, the prospects for catalytic application of ceria-based materials described in the medical related publication are hardly be evaluated, since these works do not provide the textural parameters of the synthesized materials, such as the specific surface area, pore size distribution, etc.

## Figures and Tables

**Figure 1 molecules-25-04242-f001:**
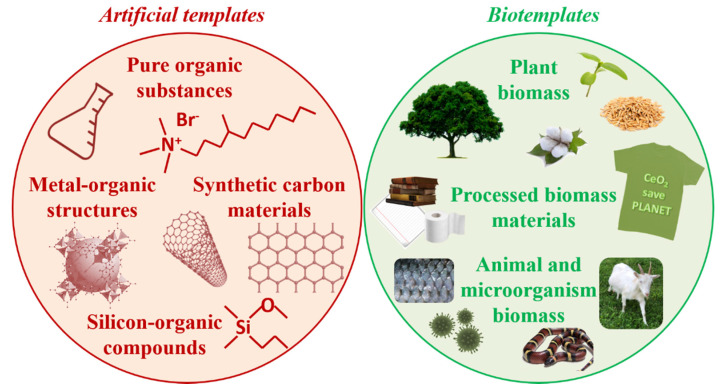
Schematic representation of the most common groups of artificial and biological materials used as templates for synthesis of ceria-based catalysts.

**Figure 2 molecules-25-04242-f002:**
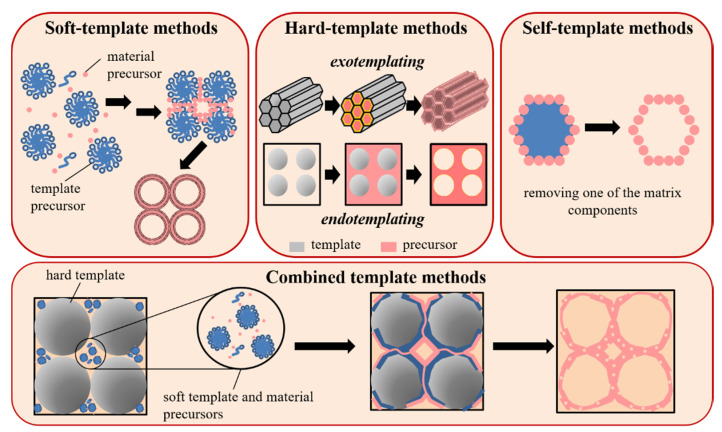
Classification of template synthesis methods by type of artificial template.

**Figure 3 molecules-25-04242-f003:**
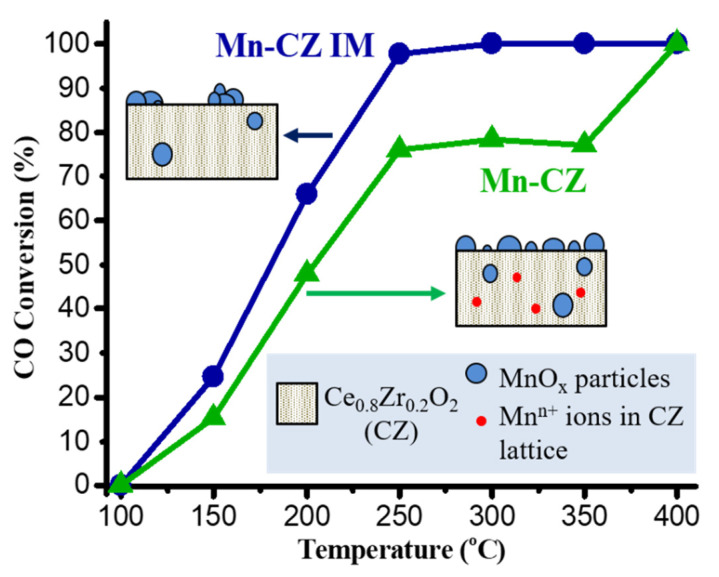
CO oxidation over Mn-CZ prepared by one-pot EISA method using CTAB template, and Mn-CZ IM, where MnO_x_ was supported on CZ by impregnation (adapted from [31]).

**Figure 4 molecules-25-04242-f004:**
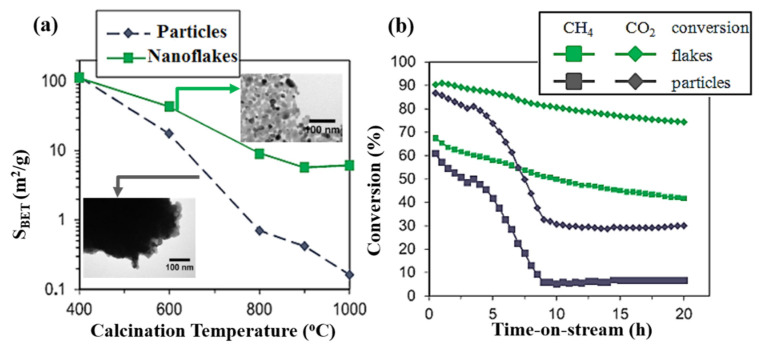
(**a**) S_BET_ vs. calcination temperature for GO-templated ceria flakes and non-templated ceria particles; (**b**) long-term catalytic activity of Ni-loaded ceria flakes and particles at 800 °C in the dry reforming of methane, both CH_4_ and CO_2_ conversions are shown (adapted from [45]).

**Figure 5 molecules-25-04242-f005:**
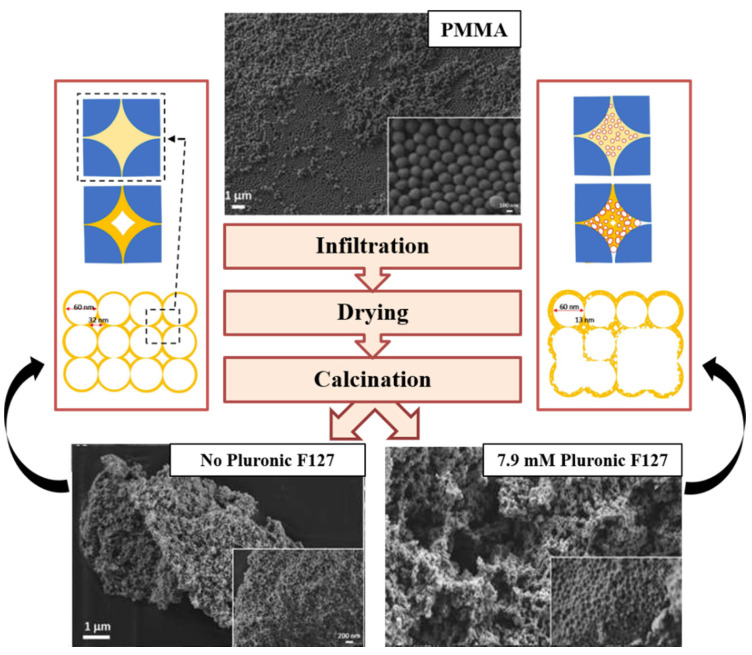
FESEM images of 3DOM catalysts prepared with PMMA hard template with or without the addition of Pluronic F127 (adapted from [65]).

**Figure 6 molecules-25-04242-f006:**
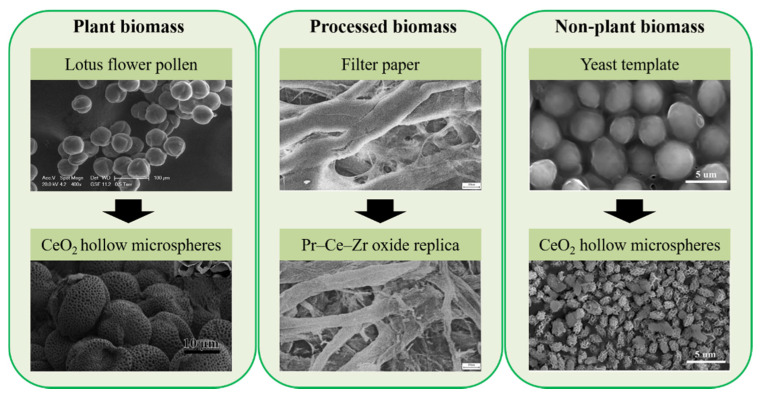
Classification of biological templates (adapted from [52,95,96]).

**Figure 7 molecules-25-04242-f007:**
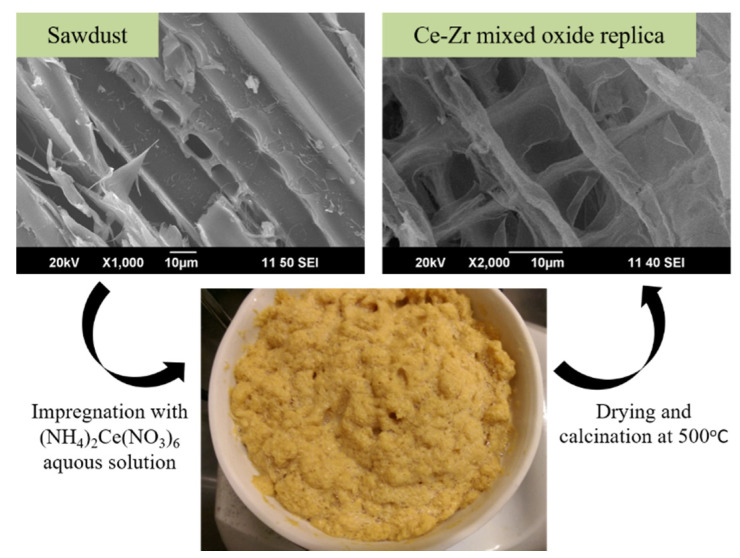
Biomorphic CeZrO_x_ reproduces well the morphology of biotemplate: SEM images of pine sawdust and CeZrO_x_ produced using this sawdust as biotemplate.

**Figure 8 molecules-25-04242-f008:**
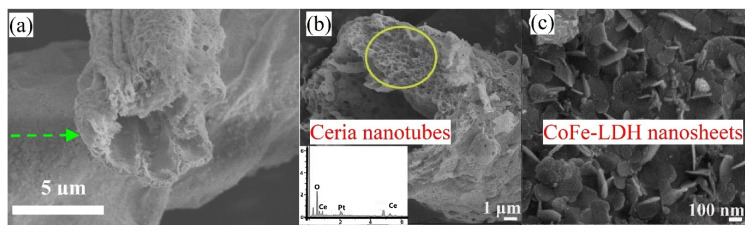
SEM image of (**a**) CeO_2_-ZnO hollow fibers (adapted from [90]); (**b**) kapok fibers templated CeO_2_ [100]; (**c**) CoFe-LDH/CeO_2_ composite (adapted from [100]).

**Table 1 molecules-25-04242-t001:** Catalytic applications of ceria-based materials produced using template methods. Explanation of abbreviations can be found in the text of the article.

Catalytic Process	Composition of the Catalyst	Template	Reference
CO oxidation	CeZrO_2_, MnO_x_-CeZrO_2_	CTAB	[31]
	CeZrO_2_, CuO_x_-CeZrO_2_	CTAB, pine sawdust	[50]
	CeSnO_2_, CuO_x_-CeSnO_2_	CTAB, Pluronic P123	[79]
	CeO_2_	Carbon nanotubes	[44]
	CeO_2_	Graphene oxide	[45]
	Cu-CeO_2_ nanosheets	Graphene oxide	[46]
	CePrO_2_	PMMA	[84]
	MnO_x_/CeO_2_	Carbon spheres	[64]
	Au/HBT-CeO_2_	SiO_2_ spheres + Ce(OH)CO_3_ nanorods	[37]
	CeO_2_-TiO_2_	Filter paper	[60]
	CeO_2_	Diatom frustule	[54]
	CeO_2_ hollow microspheres	Yeast	[94]
	Nanocrystalline CeO_2_	Crucian fish scales	[92]
	CuO/CeO_2_	Dextrin	[105]
	CeO_2_	Amino acids	[63]
Soot oxidation	Co-Fe/CeO_2_	CTAB	[23]
	Cu/3DOM CeO_2_	PMMA	[41]
	3DOM CeO_2_	PMMA + Pluronic F127	[65]
	CeO_2_, Co_3_O_4_, Co_3_O_4_-CeO_2_ hollow microfibers	Cotton	[91]
	CuO/CeZrO_x_	Loofa sponge	[102]
Oxidation and destruction of organic compounds and dyes	CeZrO_x_	CTAB	[22]
	CuO-MnOx-CeO_2_	CTAB	[26]
	CeO_2_-SiO_2_	CTAB	[28]
	CeO_2_-TiO_2_	CTAB	[73]
	CeO_2_	Pluronic F127 + SBA-15, SBA-15	[27]
	Au/CeO_2_	SBA-15	[39]
	Fibrous CeO_2_	Cellulose fibers	[61,98]
	CeO_2_	Microcrystalline cellulose	[59]
	CeO_2_ powder	Clover stems	[51]
	Au/CeO_2_	Bovine serum albumin	[56]
	M/CeO_2_, where M = Pd or Pt	NP-5 (polyethylene glycol mono-4-nonylphenyl ether)	[36]
Methane reforming, CO methanation	Ni/CeO_2_	CTAB	[22,33]
	NiO/CeO_2_-ZrO_2_	CTAB	[32]
	NiO/CeO_2_-ZrO_2_	Pluronic F123	[77]
	NiO/CeO_2_	SBA-15	[41]
	NiO/CeO_2_	g-C_3_N_4_	[47]
	NiO/CeO_2_	Graphene oxide	[45]
Water-gas shift reaction	CeO_2_	Al foam, Al sponge	[88]
SCR of NO_x_	NbO_x_@CeO_2_ nanotubes	Pluronic F123	[78]
CO_2_ reduction to CO	CeO_2_	Eastern white pine wood	[48]
Autothermal ethanol reforming	Ni/Ce(M)O_2_, where M = La, Mg, Gd	Pluronic F127	[76]
Dimethyl carbonate synthesis	CeO_2_	CTAB	[29]
	CeZrO_x_	Pluronic F-127 + spherical activated carbon	[66]
Hydrodesulphurization	CeO_2_/3DOM SiO_2_	KIT-6 + Pluronic 123	[34]
Gas sensors	ZnO-CeO_2_	Cotton	[90]
	CoFe-LDH/CeO_2_ LDH = Layered double hydroxide	Kapok fiber	[100]
Fuel-cell catalysis	CeZrO_x_, Ni/CeZrO_x_	Pluronic F123	[80]
	NiO/CeSmO_x_	NiO	[87]
	Co_3_O_4_-CeO_2_/graphene	Rape flower stem	[103]
Photocatalysis	Au@CeO_2_	CTAB	[74]
	Fibrous CeO_2_	Cellulose fibers	[98]
	ZnO/CeO_2_ nanowires	Cellulosic fibrils (banana pseudo stem)	[99]
	3DOM CeO_2_	Cork	[57]
	Microspheric N-doped CeO_2_	Lotus pollen	[52]
	Co_3_O_4_/CeO_2_	Lotus pollen	[53]
	Nanosquared CeO_2/_RGO (reduced graphene oxide)	Rape flower stem	[61]
	CeO_2_ and Fe-CeO_2_ hollow microspheres	Yeast	[95]
	CeO_2_ nanoparticles	Chitosan	[110,111]
	CeO_2_ nanoparticles	*Eucalyptus globulus* leaf extract	[113]

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
