# Peer review of "Template Synthesis of Porous Ceria-Based Catalysts for Environmental Application"

_molecules, 2020, doi:10.3390/molecules25184242_

Round 1

Reviewer 1 Report

The review by Lokteva et al summarizes recent work on the field of templated preparation of cerium oxide and related materials and their catalytic applications in biomorphic ceria-based catalytic systems. The catalyst structure-environmental application performance relationships and observed trends regarding to their catalytic efficiency affected by different templates. By means of systematic reviews on the influence of the template methods over the catalytic activity and stereochemistry, this review will be very useful for the further development of new the main areas of environmental catalysis in the future. Overall speaking, the manuscript is a well-written and properly organized paper, which will attract a wide readership from different scientific communities. As a result, I would like to recommend this manuscript to be published in molecules after the following minor issues being addressed.

  1. The size of the pore and surface area is very important for porous materials. More systematic discussions are encouraged to enhance the strength of this review.

  1. Several Figures are not clear in the present version and should be corrected to high-resolution images in the revised manuscript.

Author Response

  1. The size of the pore and surface area is very important for porous materials. More systematic discussions are encouraged to enhance the strength of this review.
    Answer: We added several points concerning pore size and surface area issues into the Section 5. Conclusions and Perspectives. They are marked yellow in the revised version of manuscript. In particular,  In lines 1073-1079: Artificial templates have a constant composition, and therefore the ceramic materials obtained on their basis have reproducible properties. However, the pore shape and their distribution in the material are difficult to control, especially in the case of soft templates, and changing the type of template even within one family requires serious and unobvious changes in the synthesis technique. In contrast, replacing one template of plant origin with another does not require a change in the synthesis procedure, but allows one to vary the size and shape of pores in the final ceramics within a wide range. In lines 1105-1110: Since the specific surface area and pore size are critical characteristics of catalysts, it is interesting to trace the effect of the type of the chosen template on them. Biological templates of plant origin, as a rule, are characterized by the presence of large pores; therefore, ceramic materials obtained on their basis have a smaller surface and a larger pore size in comparison with those obtained using artificial templates. For this reason, biological templates are advantageously selected for preparing catalysts intended for processing relatively large species, for example, soot particles. In lines 1112-1117: Biotemplates are renewable resources, so biotemplating is beneficial and promising from the point of view of current global trends towards the preservation of non-renewable raw materials. A wide variety of morphologies of biomaterials, as well as pore sizes and arrangement, provide tools for adjusting texture properties. However, the dependence of the biomaterial composition on natural conditions makes it difficult to obtain ceramics with reproducible properties.
    Also, several other changes were made in this section, they are also marked in yellow.
  2.  Several Figures are not clear in the present version and should be corrected to high-resolution images in the revised manuscript.
    Answer: We improved the resolution of all images from 300 to 600 dpi and partially changed images in Fig. 8 by high-resolution images taken from the site of the corresponding Journal.

Reviewer 2 Report

The contents presented in review paper from the title “Template synthesis of porous ceria-based catalysts for environmental application” should be attractive to the readers of the Molecules journal. The authors successfully done investigation which includes summarizing and analysis the most recent reports published in the last decade on the template synthesis and catalytic applications of ceria-based materials with defined porous structure.

However, based on the presented discussions only three references of the authors themselves were noticed, which are, among other things, common. It would be good for the authors to supplement the work with their own research on the topic they covered, i.e. what is the contribution of their personal research from the aspect of innovation and contributions related to this topic.

As well, I would recommend the authors to throw out Figure 9 and present the catalytic applications of templated ceria-based materials in a different way.

I recommend acceptance for publication after minor modifications.

Author Response

  1. However, based on the presented discussions only three references of the authors themselves were noticed, which are, among other things, common. It would be good for the authors to supplement the work with their own research on the topic they covered, i.e. what is the contribution of their personal research from the aspect of innovation and contributions related to this topic.                                                                                Answer: The authors started work in this field not long ago. Until now, only a few papers in this area have been published. We believe that this is not an obstacle to summarizing and discussing works published by other authors in this area.                                                                             To strengthen the discussion of the author’s work, we added the discussion of one more work of the authors published this year, and several remarks concerning the other references already cited in the original version:   Lines 212-213: Thus, it was clearly demonstrated in our work that the method of adding the third component to cerium-zirconium systems has a significant effect on the catalytic properties.                                       Lines 372-383: Tin oxide is another promising modifier due to the ability to form solid solutions with cerium oxide and the relatively low cost as compared to zirconia. A CO oxidation activity of CeSnOx (CS) and CuOx/CeSnOx (Cu-CS) catalysts prepared by CTAB or Pluronic P123 was compared in our scientific group [79]. Сatalytic properties of prepared systems strongly depend on the template nature and copper modification technique because these parameters determine degree of interaction between different components in Cu–Ce–Sn oxide systems. The combination of CTAB-templated method and “one-pot” copper addition technique led to the more uniform distribution and partial incorporation of copper ions into the CS lattice, which provided high oxygen mobility and defectiveness. These facts explain why Cu-CS CTAB
    sample (SBET = 84±8 m2/g) exhibited excellent catalytic properties over the entire temperature range studied. In contrast, Pluronic 123-templated counterpart (SBET = 96±9 m2/g) showed 40–60% conversion of CO only at relatively low temperatures, and it was less effective in the high-temperature range.
    Lines 818-820: Earlier in the literature, we did not see any mention of the preparation of a cerium-zirconium catalyst using a biotemplate, which is more efficient in the oxidation of CO than an analogue obtained using CTAB as an artificial template.                                                                     Line 1083: However, as it was found in our work [33], when modified ceria materials are synthesized, the stability of the respective complexes should be carefully evaluated because it can crucially affect the porosity of synthesized oxides; thus, Mn addition in the EISA method using CTAB template deteriorates texture properties.
  2. As well, I would recommend the authors to throw out Figure 9 and present the catalytic applications of templated ceria-based materials in a different way.                                                                                            Answer: In accordance with this remark, we deleted Figure 9 and replaced it with Table 1, presenting the list of catalytic applications of templated materials based on ceria. The table is organized as follows:
    Catalytic process      Composition of the catalyst       Template    Reference Changes in the text of this section, caused by the replacement of the Figure 9 with a table, are highlighted in the text with a yellow marker.